# A heterotrimeric complex of *Toxoplasma* proteins promotes parasite survival in interferon gamma-stimulated human cells

Eloise J. Lockyer[1], Francesca Torelli[1], Simon Butterworth[1], Ok-Ryul Song[2], Steven Howell[3], Anne Weston[4], Philip East[5], Moritz Treeck[1,6]*

**1** Signalling in Apicomplexan Parasites Laboratory, The Francis Crick Institute, London, United Kingdom, **2** High-Throughput Screening Science Technology Platform, The Francis Crick Institute, London, United Kingdom, **3** Proteomics Science Technology Platform, The Francis Crick Institute, London, United Kingdom, **4** Electron Microscopy Science Technology Platform, The Francis Crick Institute, London, United Kingdom, **5** Bioinformatics and Biostatistics Science Technology Platform, The Francis Crick Institute, London, United Kingdom, **6** Cell Biology of Host-Pathogen Interaction Laboratory, Instituto Gulbenkian Ciência, Oeiras, Portugal

* moritz.treeck@crick.ac.uk, mtreeck@igc.pt

**Data Availability Statement:** All relevant data are within the paper and its Supporting information files. Proteomics raw data are uploaded into the

## Abstract

*Toxoplasma gondii* secretes protein effectors to subvert the human immune system sufficiently to establish a chronic infection. Relative to murine infections, little is known about which parasite effectors disarm human immune responses. Here, we used targeted CRISPR screening to identify secreted protein effectors required for parasite survival in IFNγ-activated human cells. Independent screens were carried out using 2 *Toxoplasma* strains that differ in virulence in mice, leading to the identification of effectors required for survival in IFNγ-activated human cells. We identify the secreted protein GRA57 and 2 other proteins, GRA70 and GRA71, that together form a complex which enhances the ability of parasites to persist in IFNγ-activated human foreskin fibroblasts (HFFs). Components of the protein machinery required for export of *Toxoplasma* proteins into the host cell were also found to be important for parasite resistance to IFNγ in human cells, but these export components function independently of the identified protein complex. Host-mediated ubiquitination of the parasite vacuole has previously been associated with increased parasite clearance from human cells, but we find that vacuoles from GRA57, GRA70, and GRA71 knockout strains are surprisingly less ubiquitinated by the host cell. We hypothesise that this is likely a secondary consequence of deletion of the complex, unlinked to the IFNγ resistance mediated by these effectors.

## Introduction

*Toxoplasma gondii* is an obligate intracellular parasite that can infect any nucleated cell of virtually any warm-blooded animal. With an estimated human worldwide seroprevalence of 30%, it is also likely one of the most prevalent human protozoan parasites [1]. Though *Toxoplasma* can cause disease in immunocompromised patients or following congenital transmission,

PRIDE repository (PXD041352), mRNAseq raw data files have been submitted to GEO with the accession number (GSE230866).

**Funding:** The work of EJL, FT, SB and MT was supported by the Francis Crick Institute which receives its core funding from Cancer Research UK (CC2133), the UK Medical Research Council (CC2133), and the Wellcome Trust (CC2133). FT received funding from the Deutsche Forschungsgemeinschaft (TO 1349/1-1). OS, SH, AW and PE were supported by the Science Technology Platforms at the Francis Crick Institute, which receive funding from Cancer Research UK (CC0199), the UK Medical Research Council (CC0199), and the Wellcome Trust (CC0199). The funders play no role in study design, data collection and analysis, decision to publish, or preparation of the manuscript.

**Competing interests:** The authors have declared that no competing interests exist.

**Abbreviations:** BMDM, bone marrow-derived macrophage; BSA, bovine serum albumin; FDR, false detection rate; GBP1, guanylate-binding protein 1; gDNA, genomic DNA; GEO, Gene Expression Omnibus; HFF, human foreskin fibroblast; HUVEC, human umbilical vein endothelial cell; IRG, immunity-related GTPase; ISG, IFN-stimulated gene; IVN, intravacuolar network; LC, liquid chromatography; LFQ, label-free quantification; MAD, median absolute deviation; MEF, mouse embryonic fibroblast; PFA, paraformaldehyde; PV, parasitophorous vacuole; PVM, parasitophorous vacuole membrane; sgRNA, single- guide RNA; TEM, transmission electron microscopy; TFA, trifluoroacetic acid; TMD, transmembrane domain.

most immunocompetent individuals control the infection through the combined action of the adaptive and innate immune systems.

The *Toxoplasma gondii* lifecycle comprises 2 distinct phases, a sexual stage that takes place exclusively in the definitive feline host and an asexual stage that can occur in a very broad range of intermediate hosts [2]. The evolutionary importance of an intermediate host is intrinsically linked to the frequency with which it promotes transmission of *Toxoplasma* parasites back to the felid definitive host [3]. Rodent species and other small warm-blooded animals that are prey to felines are therefore important natural intermediate hosts for *Toxoplasma*, but as humans are very rarely natural feline prey, they are considered "accidental" intermediate hosts.

Once within an intermediate host, *Toxoplasma* actively invades host cells and resides in a host-derived membrane-bound vacuole known as the parasitophorous vacuole (PV). In mice and humans, host cell-autonomous immunity to *Toxoplasma* is mediated by the type II interferon IFNγ. IFNγ stimulation leads to the up-regulation of hundreds of IFN-stimulated genes (ISGs) through STAT1-induced transcription of gamma-activated sequence elements [4,5]. To counteract the IFNγ response, *Toxoplasma* secretes protein effectors from specialised rhoptry and dense granule organelles. Dense granule proteins (GRAs) and rhoptry proteins (ROPs) transform the host cell by altering gene transcription and host metabolism, mediating the acquisition of nutrients, removing host cell factors from the PV membrane (PVM) and targeting proinflammatory signalling pathways (reviewed in [6–11]). Global mapping of the *Toxoplasma* proteome using hyperplexed localisation of organelle proteins by isotope tagging (hyperLOPIT) has predicted the existence of at least 124 GRAs and 106 ROPs [12]. Sequence polymorphisms and differential expression of secreted effectors across the 3 main clonal lineages of *Toxoplasma* (types I–III) have been shown to determine differences in strain virulence [13,14]; however, the vast majority of ROPs and GRAs remain uncharacterised.

GRAs can be translocated across the parasite plasma membrane and PVM into the host cell cytoplasm via the MYR complex, a multi-protein putative translocon. To date, 8 proteins have been identified as necessary for protein translocation: the putative MYR translocon components; MYR1, MYR2, MYR3, MYR4, and ROP17 [15–18], as well as GRA44, GRA45, and ASP5 [18,19]. The secretion of MYR-dependent effectors is responsible for the vast majority of host cell transcriptome changes induced by *Toxoplasma* infection in both human and mouse cells [20,21]. MYR1 knockouts display an expected reduction in virulence *in vivo*, but this phenotype is lost in pooled knockout CRISPR screens, suggesting MYR1-dependent effectors may exert paracrine effects at the site of infection [22]. How the MYR proteins directly facilitate protein translocation is not yet understood, and while MYR1 and MYR3 stably associate *in vitro* [16], it is not yet clear whether all MYRs form a single complex required for protein translocation.

Efforts to characterise the function of *Toxoplasma*-secreted effectors historically centred on their role in the murine host, leading to the discovery of ROP18, ROP5, and ROP17 [23–26]. In mice, these effectors act cooperatively to prevent the loading of the immunity-related GTPase (IRG) family of large GTPases [27–32], which are strongly induced by IFNγ in multiple mouse cell types [33,34]. IRG proteins load sequentially onto the PVM and cooperatively oligomerise to vesiculate and rupture the membrane, leading to parasite clearance and necrotic host cell death [33,35,36]. The allelic combination of ROP18/ROP5/ROP17 in each parasite strain determines the strain-specific virulence of *Toxoplasma* in mice [37]. While mice have a dramatically expanded family of 23 IRGs, IRGs are largely absent in humans having been mostly lost in the primate lineage prior to the evolution of monkeys [38]. As a result, ROP18/ROP17/ROP5 do not function similarly to resist vacuole clearance in human cells [31].

In humans, the IFNγ-induced responses to *Toxoplasma* are highly cell-type specific. These mechanisms include autophagic clearance [39], endo-lysosomal destruction [40,41], nutrient deprivation [42,43], or induction of host cell death [44]. In human macrophages, the p65 GTPase guanylate-binding protein 1 (GBP1) has been demonstrated to mediate killing of *Toxoplasma* through recruitment to and disruption of the PVM, leading to host cell apoptosis [45].

In multiple mouse and human cell types, IFNγ-driven ubiquitination of the PVM by host E3 ligases serves as an initial marker for eventual parasite clearance. In mouse embryonic fibroblasts (MEFs) and bone marrow-derived macrophages (BMDMs), TRAF6-mediated ubiquitination of type II and III vacuoles enhances the recruitment of GBPs, resulting in rupture of the PVM [41,46]. The E3 ligase TRIM21 has also been shown to promote clearance of type II and III strains *in vivo* [47] and ubiquitinates type III vacuoles in human foreskin fibroblasts (HFFs) resulting in parasite growth restriction [48]. In human umbilical vein endothelial cells (HUVECs), K63-linked ubiquitination initiates a cascade of autophagy marker recruitment that culminates in acidification of the parasite vacuole [40], while in HeLa cells ubiquitination instead leads to recruitment of LC3 and stunting of parasite growth [39]. Compared to type II and III parasites, type I parasites are more resistant to ubiquitination and clearance in both HUVECs and HeLa cells. Recent evidence suggests a major function for the E3 ligase RNF213 in ubiquitination of the PVM and enhancing parasite clearance, in multiple human cell types and in a strain-independent manner [49,50]. The importance of each E3 ligase and ubiquitin linkage therefore depends on the exact combination of host species, parasite strain, and cell type studied, but nevertheless ubiquitination has emerged as a key process in the regulation of *Toxoplasma* clearance. The *Toxoplasma*-derived targets that are recognised by these host E3 ligases remain to be determined.

*Toxoplasma* effectors that have been described to counteract IFNγ in human cells include the host transcriptional repressor IST, which blocks IFNγ-stimulated gene transcription [51,52]. However, IST can only prevent clearance in cells that have not been pre-stimulated with IFNγ, a condition that may only be met in the early stages of infection. Another effector is NSM, which blocks host cell necroptosis during the bradyzoite stages of infection [53]. More recently, we have shown that ROP1 counteracts IFNγ immune responses in both murine and human macrophages [54]. Finally, in human THP-1-derived macrophages, deletion of the chaperone protein GRA45 increases the sensitivity of parasites to IFNγ-mediated growth inhibition [55]. There therefore remains a significant gap in our understanding of which effectors mediate *Toxoplasma* virulence in humans [3,56].

To address this gap, we performed targeted CRISPR screening of the *Toxoplasma* "secretome" [12] during infection of unstimulated and IFNγ-stimulated HFFs, using our previously described CRISPR platform [22]. Independent experiments were performed in type I (RH) and type II (PRU) parasite strains, which allowed identification of both strain-dependent and independent effectors. We found that GRA57 is a strain-independent effector that protects *Toxoplasma* from IFNγ-mediated vacuole clearance in HFFs. GRA57 was found to interact with 2 other dense granule proteins, GRA70 (TGME49_249990) and GRA71 (TGME49_309600), which resist IFNγ-mediated vacuole clearance to a similar degree as GRA57, indicating that the 3 proteins function in the same pathway, possibly as a complex. Deletion of any member of this trio results in reduced PVM ubiquitination in HFFs. We also found 2 components of the MYR translocon, MYR1 and MYR3, contribute to IFNγ resistance in HFFs, though we expect that one or several unidentified MYR-dependent effectors play an additional significant role. As GRA57 does not impact GRA export, and MYR1 and MYR3 do not display an ubiquitination phenotype, we conclude that GRA57 and the MYR proteins function independently of each other. These data suggest that a novel MYR-independent

trimeric complex of dense granule proteins localised within the PV contribute to resisting IFNγ-induced vacuole clearance in HFFs.

## Results

### CRISPR screening the *Toxoplasma* secretome in IFNγ-activated human cells

To screen for effector proteins that promote *Toxoplasma* survival in IFNγ-activated human cells, we used a previously established *in vivo* CRISPR screening platform [22] with a guide library targeting 253 predicted or validated dense granule and rhoptry proteins. To allow for identification of strain-specific and strain-independent effectors, we generated type I and type II CRISPR knockout parasite pools by transfecting this library into RHΔHXGPRT and PRUΔHXGPRT strains. The vector library contains an HXGPRT resistance cassette, allowing for positive selection of integrants using mycophenolic acid and xanthine (M/X). Transfected parasites were selected with M/X in HFFs for 8 days to generate a pooled mutant parasite population. Pooled mutant parasites were then used to infect HFFs that were either untreated or pre-stimulated with IFNγ, for 2 lytic cycles (96 h) (Fig 1A). The RH and PRU mutant populations were both highly restricted in IFNγ-stimulated HFFs after 48 h of infection.

To identify genes that confer a growth benefit for *Toxoplasma* in HFFs in the presence of IFNγ, we measured the relative abundance of guide RNAs (gRNAs) targeting each gene in the *Toxoplasma* mutant library before and after selection in IFNγ-treated or untreated HFFs. Sequencing read counts were used to calculate the median log2 fold change (L2FC) of guides targeting each gene between conditions and matched gene concordance/discordance (DISCO) to compare gene phenotypes between treatment conditions. To assess which effectors contributed predominantly to survival of IFNγ responses in HFFs, we calculated the L2FC for each gene between IFNγ-stimulated and untreated cells after 2 rounds of selection, then compared this to *in vitro* growth phenotypes using the L2FC between untreated cells and the starting parasite inoculum. We selected effectors with IFNγ-specific phenotypes using a L2FC cutoff of < -1 in IFNγ-activated HFFs relative to unstimulated HFFs and > -1 in unstimulated HFFs relative to the inoculum.

Results from both the RH and PRU screens showed that ROP18 and GRA12–2 major virulence factors in murine cells and *in vivo* ([23,31,57,58])—have no major impact on parasite survival in HFFs (Fig 1B and 1C). In the RH screen, effectors with the highest IFNγ phenotype were CYP18, a protein with a proinflammatory effect in macrophages [59]; MYR3, a component of the dense granule export translocon [16]; GRA57, a cyst-localising protein [60]; and TGME49_309600, a hypothetical dense granule protein (Fig 1B). In the PRU screen, the highest scoring effectors were GRA35, an inducer of pyroptosis in rat macrophages [61]; GRA25, a major virulence factor in mice [62]; MYR3, TGME49_249990, a predicted dense granule protein; and GRA57 (Fig 1C).

To identify strain-independent effectors contributing specifically to survival of HFF IFNγ responses, we compared genes with strong IFNγ phenotype scores (L2FC < -1) in the RH and PRU screens, while excluding those with strong phenotype scores (L2FC < -1) in unstimulated HFFs (Fig 1D and 1E and S1 Data). For both screens, 3 components of the MYR translocon—MYR1, MYR3, and ROP17—were among the genes with the highest phenotype scores for survival in IFNγ-activated HFFs. As the export of many dense granule proteins into the host cell has been shown to be MYR-dependent, it is likely that the role of MYR components in IFNγ resistance results from the pleiotropic effect of abrogating dense granule export. We instead focused our efforts on the further characterisation of GRA57, which showed the second strongest phenotype in both screens and had no previously known function.

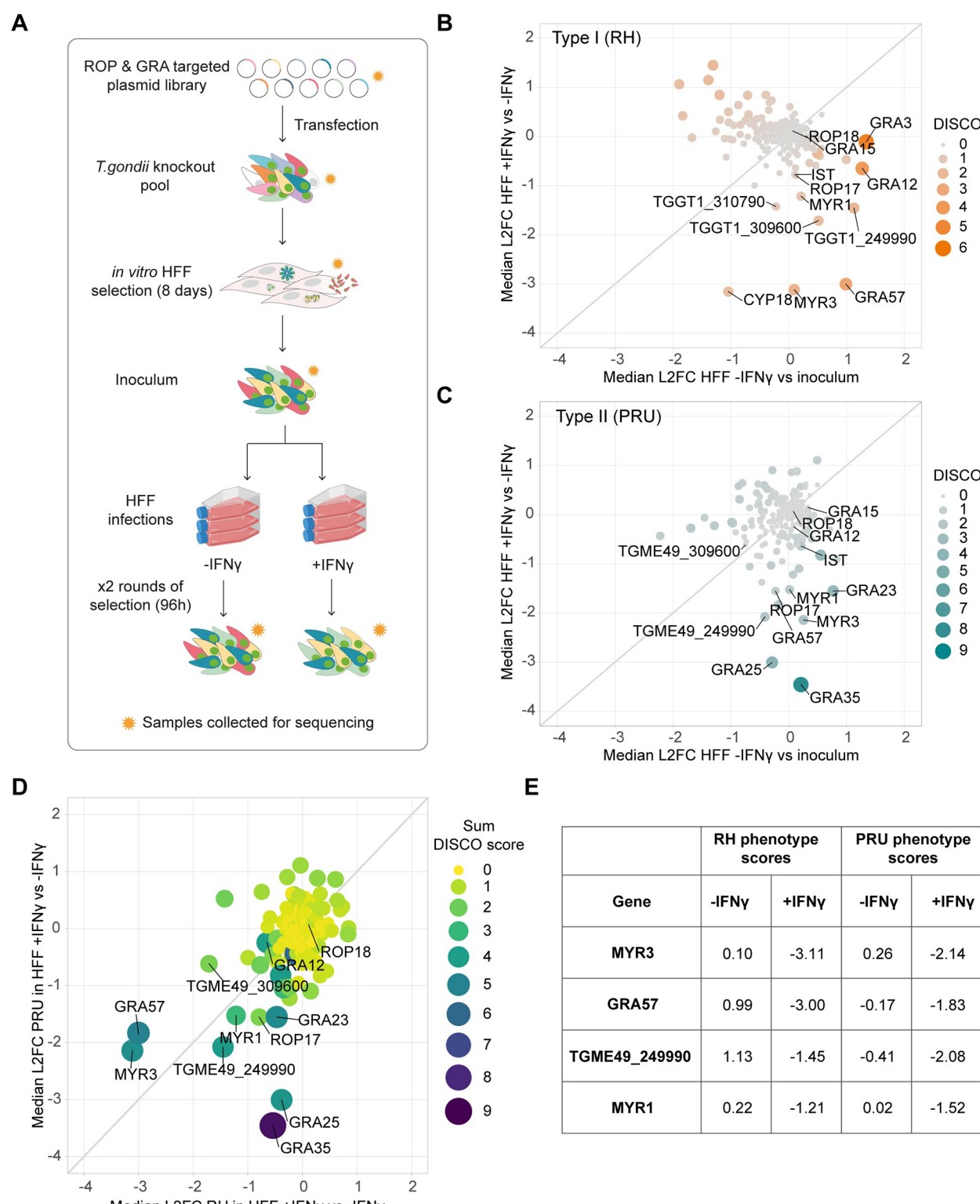

**Fig 1. CRISPR screening the *Toxoplasma* secretome to identify novel effectors in IFNγ-activated HFFs. (A)** Schematic of experimental design. A pCas9-T2A-HXGPRT-sgRNA vector library designed against predicted secreted effectors was transfected into RH or PRUΔHXGPRT to generate a mutant parasite pool. Transfectants were grown under M/X selection in HFFs for 8 days, and then combined to generate the inoculum. HFFs were infected in triplicate, in the presence or absence of 24 h pre-stimulation with 100 U/ml IFNγ. After 48 h infection, a subset of parasites from each condition were retrieved and used for a second round of infection for a further 48 h. Subsets of parasite pools were taken for gDNA extraction

and sequencing at each stage (indicated by orange stars) to determine the relative abundance of guides. (**B and C**) Discordance plots showing median log2 fold changes (L2FCs) for *Toxoplasma* effector genes in (**B**) RH background and (**C**) PRU background. L2FCs for growth in IFNγ-activated HFFs are plotted against control L2FCs for growth in unstimulated cells. Colour and size of data points indicate strength of discordance score.(**D**) Correlation between RH and PRU screens in IFNγ-activated HFFs. Genes with L2FCs <-1 for growth in unstimulated HFFs in either screen are excluded. (**E**) Table of genes with L2FCs <-1 for growth in IFNγ-activated HFFs in both RH and PRU backgrounds. Source data for B–E can be found in S1 Data. gDNA, genomic DNA; HFF, human foreskin fibroblast; M/X, mycophenolic acid and xanthine.

## GRA57 is an intravacuolar protein required for *Toxoplasma* survival of IFNγ responses in HFFs

To investigate the function of GRA57, we generated a knockout line in the type I RH strain via Cas9-mediated gene disruption and integration of an mCherry-T2A-HXGPRT repair cassette. We complemented the GRA57 coding sequence with a C-terminal HA tag back into this line at the non-essential UPRT locus (S1A and S1B Fig). This allowed us to confirm the localisation of GRA57-HA described in [60] via immunofluorescence, showing that GRA57 resides within the vacuole (Fig 2A), and partially co-localises with GRA3 (S3 Fig), a marker of the PVM [63]. GRA57 is a relatively large dense granule protein predicted to be 246 kDa, which we confirmed via western blot (Fig 2B).

To verify the phenotype observed in the CRISPR screens, we assessed the survival of the RHΔGRA57 and RHΔGRA57::GRA57-HA strains in IFNγ-activated HFFs. HFFs were pre-stimulated with 100 U/ml IFNγ for 24 h prior to infection with the mCherry expressing parasite lines or left untreated as a control. As a wild-type control, we used a parental strain where an mCherry-T2A-HXGPRT cassette was introduced into the UPRT locus. The total mCherry signal area was measured from microscopy images taken on a Cytation plate imager as a read-out for *Toxoplasma* survival, and the percentage survival was calculated as a proportion of the area in unstimulated versus IFNγ-stimulated cells. We also generated an RHΔMYR3 strain and included this alongside RHΔMYR1 given the strong phenotype observed for these effectors in the pooled CRISPR screens.

In IFNγ-stimulated HFFs, RHΔGRA57 parasites displayed a relative decrease in survival of 39% compared to RHΔUPRT controls (Fig 2C). This decrease in survival was rescued in the complemented line, confirming that the absence of GRA57 renders parasites more susceptible to IFNγ restriction in HFFs. RHΔMYR3 and RHΔMYR1 were more highly restricted, with relative reductions in survival of 68% and 84%, respectively. This indicates that while GRA57 plays an important role in resisting host cell restriction, it is unlikely to have a major influence on MYR-dependent protein export. This is further investigated later in this manuscript.

Since GRA57 has not previously been identified as an important effector for survival in mice [55,64], this indicates that GRA57 plays a role in only some species or specific cell types. To verify this, we performed restriction assays in IFNγ-activated MEFs as an equivalent murine cell type. Survival levels for all parasite lines were higher in MEFs (91% for RHΔUPRT) than in HFFs (46% for RHΔUPRT), suggesting that HFFs restrict type I parasites more effectively than MEFs, at least under the conditions tested (Fig 2D). This is in line with previously published data that shows type I parasites are highly resistant to restriction in MEFs [44].

RHΔGRA57 parasites displayed a relative decrease in survival of 11%; however, this was not a significant change, indicating that GRA57 is not important for survival in MEFs. In contrast, RHΔMYR3 parasites had a significant relative reduction in survival of 31%, with RHΔMYR1 showing a similar, but not statistically significant trend of 30% relative reduction, indicating that preventing effector export has a major impact on parasite survival regardless of host species. Overall, this data verifies that GRA57 and components of the MYR translocon are novel *Toxoplasma* survival factors that contribute to parasite resistance to IFNγ in HFFs.

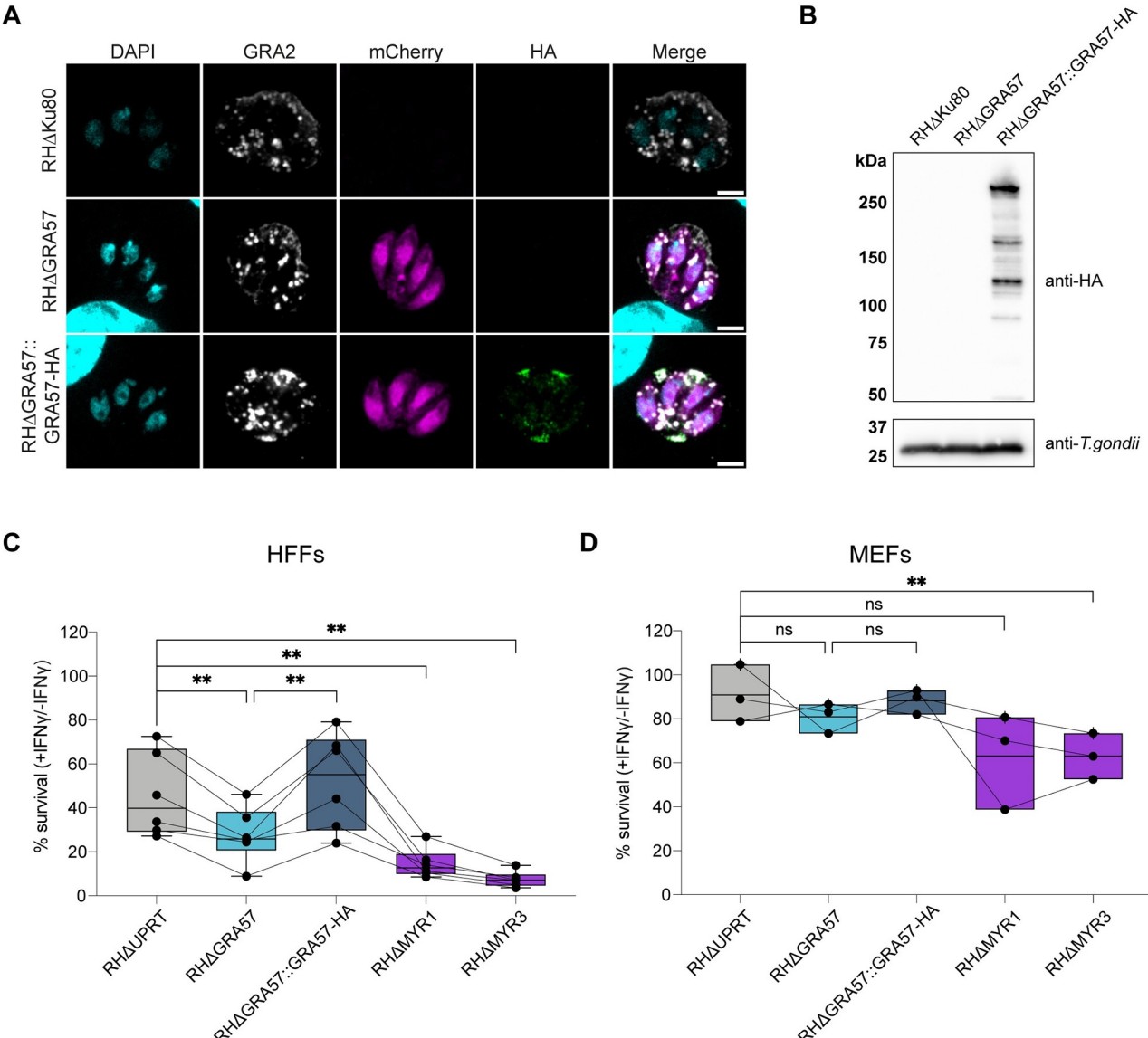

**Fig 2. GRA57 is an intravacuolar protein that contributes to parasite survival of IFNγ in HFFs. (A)** IFA of RHΔGRA57 and RHΔGRA57::
GRA57-HA lines generated for this study. HA-tagged GRA57 co-localises with GRA2—a marker of the IVN. Scale bar = 3 μm. **(B)** Western blot
analysis of GRA57-HA shows it is a 250 kDa protein that is relatively highly expressed. **(C and D)** Live restriction assays in **(C)** HFFs and **(D)** MEFs
using mCherry fluorescence area as a readout for parasite survival. Host cells were pre-stimulated for 24 h with 100 U/ml IFNγ or left untreated then
infected in technical triplicate with the indicated parasite strains for 48 h (HFFs) or 24 h (MEFs) at an MOI of 0.3. Infected cells were then imaged live
on a Cytation plate reader. Total mCherry signal area per well was measured to determine parasite growth in IFNγ-stimulated relative to unstimulated
cells. Data displayed as median survival with individual biological replicates overlayed. *p*-values were calculated by paired two-sided *t* test. **, *p* < 0.01;
ns, not significant. Source data for C and D can be found in S3 Data. HFF, human foreskin fibroblast; MEF, mouse embryonic fibroblast.

## GRA57 forms a complex with TGGT1_249990 and TGGT1_309600

To identify interaction partners of GRA57 during infection of HFFs, which could indicate how
GRA57 functions to protect the parasite, we performed co-immunoprecipitation (IP) experi-
ments with an endogenously tagged RHGRA57-HA strain (S2 Fig). To enable detection of
protein–protein interactions specific to activated cells, HFFs were pre-stimulated before infec-
tion with 2.5 U/ml IFNγ for 6 h, which we found was sufficient to induce IFNγ responses

while retaining host cell viability. HFFs were infected for 24 h with either RHΔKu80 or RHGRA57-HA parasites prior to lysis. GRA57-HA was immunoprecipitated from lysates, and then co-immunoprecipitated proteins were identified by liquid chromatography (LC)-tandem mass spectrometry (MS/MS). The full MS dataset is shown in S4 Data.

GRA57 was significantly enriched in RHGRA57-HA infected samples relative to RHΔKu80 infected samples (Fig 3A). Strikingly, the 2 most strongly co-enriched proteins in the tagged samples were TGGT1_249990 and TGGT1_309600 (Fig 3A)—2 predicted dense granule proteins that also displayed IFNγ-specific phenotypes in both CRISPR screens (Fig 1D and S1 Data). The mean label-free quantification (LFQ) values detected for GRA57, TGGT1_249990, and TGGT1_309600 were similar (Fig 3B), which strongly suggests that GRA57, TGGT1_249990, and TGGT1_309600 form a heterotrimeric complex. Because both TGGT1_249990 and TGGT1_309600 are predicted dense granule proteins that interact with GRA57, they are referred to as GRA70 and GRA71, respectively, from here onwards. All 3 proteins are predicted to have transmembrane helices, with GRA57 and GRA70 also showing regions of predicted coiled-coil towards the N-terminus (Fig 3C).

To show further evidence of this complex, we determined the localisation of GRA70 via immunofluorescence. GRA70 was C-terminally tagged with a V5 epitope to validate its localisation in the PV (S2 Fig). Western blotting showed the expected size of GRA70 (93 kDa) (Fig 3D), and immunofluorescence confirmed that GRA70 is secreted into the PV (Fig 3E and 3F). GRA70 and GRA57 display similar patterns of intravacuolar localisation at 24 h post-infection, partially co-localising with the IVN marker GRA2 [65] (Fig 3F). This localisation did not change when HFFs were pretreated with IFNγ (S4 Fig). With saponin permeabilisation, we could only detect minor signal for GRA70-V5 in the vacuole despite clear signal for GRA3 (Fig 3E), a PVM marker that is partially exposed to the host cell cytosol [63]. In contrast, GRA57-HA was detectable with both saponin and triton permeabilisation (Fig 3E and 3F), suggesting the C-terminal portion of GRA57 is exposed to the host cell cytosol whereas the GRA70 C-terminus remains within the PV. This is supported by results from another group that detected GRA57, but not GRA70 or GRA71, enriched at the PVM [66]. We did not detect any signal beyond the PVM for either GRA70 or GRA57, suggesting that these effectors are not secreted into the host cell. Together, this data suggests that GRA57, GRA70, and GRA71 form a complex that is anchored in the PVM, or IVN, via GRA57.

## GRA57, GRA70, and GRA71 confer resistance to vacuole clearance in IFNγ-activated HFFs

To verify the phenotype seen in CRISPR screens for GRA70 and GRA71, we generated individual knockouts of GRA71 and GRA70 in the RH strain (S1 Fig). To assess with higher precision whether these parasites are more susceptible to IFNγ-induced growth restriction or vacuole clearance, we analysed the survival of these lines in IFNγ-activated HFFs using a high-content imaging method previously established in [54].

GRA57 knockouts displayed increased sensitivity to IFNγ in HFFs (Fig 4A and 4B), concurrent with the phenotype observed in the CRISPR screen (Fig 1B). RHΔGRA70 and RHΔGRA71 parasites were restricted to the same level as RHΔGRA57 parasites (mean relative reduction in overall parasite numbers vs. RHΔUPRT—30%, 33%, and 31%, respectively), confirming that these are also resistance factors in activated HFFs (Fig 4A). For GRA57, GRA70, and GRA71, the overall reduction in parasite survival could mainly be attributed to a reduction in the number of parasite vacuoles (Fig 4B), with no significant decrease in the size of parasite vacuoles (Fig 4C). This suggests these knockout lines are more susceptible to active clearance, early egress from the host cell [44], or increased host cell death, rather than restriction of

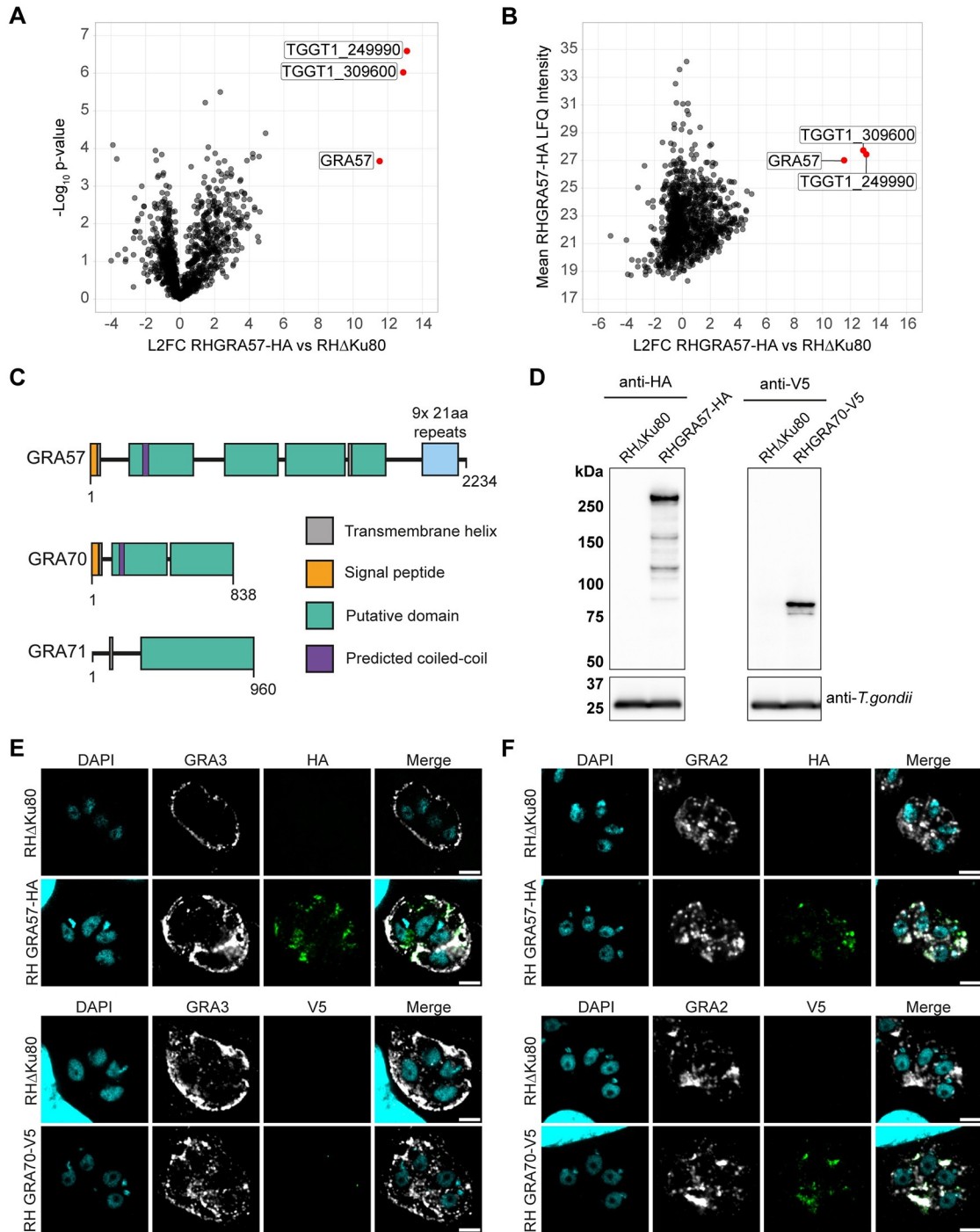

**Fig 3. CoIP of GRA57-HA shows it likely forms a complex with TGGT1_249990 and TGGT1_309600 within the PV. (A, B)** Co-immunoprecipitation of endogenously HA-tagged RHGRA57. HFFs were pre-stimulated with 2.5 U/ml IFNγ for 6 h, then infected in triplicate with either RHΔKu80 or RHGRA57-HA for a further 24 h. GRA57 was immunoprecipitated with anti-HA agarose matrix then immunoprecipitate was analysed by mass spectrometry. **(A)** Enrichment of proteins in RHGRA57-HA vs. RHΔKu80 control samples. **(B)** Mean LFQ signal intensity detected for proteins in RHGRA57-HA samples. **(C)** Schematic of putative domains in GRA57, GRA70, and GRA71, to scale. Domains and boundaries were assigned based on regions of predicted secondary structure using PsiPred. Transmembrane helices are annotated where predicted by at least 2 programmes (MEMSAT-SVM, DeepTMHMM, and CCTOP). Regions of predicted coiled-coil were determined using Waggawagga. **(D)** Western blot protein expression analysis of endogenously tagged lines. **(E)** Co-localisation of endogenously c-terminally tagged GRA57 and GRA70 with the PVM marker GRA3, which is exposed to the host cell cytosol. Samples were permeabilised with 0.1% saponin. Scale bar = 3 μm. **(F)** Co-

localisation of endogenously c-terminally tagged GRA57 and GRA70 with the IVN marker GRA2. Samples were permeabilised with 0.2% Triton-X100. Scale bar = 3 μm. Source data for A and B can be found in S4 Data. HFF, human foreskin fibroblast; IVN, intravacuolar network; LFQ, label-free quantification; PV, parasitophorous vacuole; PVM, parasitophorous vacuole membrane.

parasite growth. For the MYR component knockouts, parasites numbers were reduced as a result of a decrease in both vacuole number and size (Fig 4B and 4C), which may be expected given the combinatorial effects of blocking export of many parasite effectors. Given that deletion of all 3 components of the complex leads to a very similar reduction in resistance to vacuole clearance (mean relative reduction in vacuole number vs. RHΔUPRT-GRA57: 33%, GRA70: 34%, GRA71: 41%), we conclude that GRA57, GRA70, and GRA71 form a complex that mediates resistance to vacuole clearance in IFNγ-stimulated HFFs.

## GRA57 does not affect tryptophan metabolism, effector export, or gene expression

To determine the mechanism through which GRA57 protects parasites from IFNγ defences, we investigated the role of tryptophan metabolism in the increased susceptibility of RHΔGRA57 parasites in HFFs. Depletion of tryptophan, for which *Toxoplasma* is auxotrophic, has previously been described as an IFNγ-induced restriction mechanism in HFFs. IFNγ-induced expression of the enzyme indoleamine 2,3-dioxygenase (IDO1) results in degradation of intracellular tryptophan, limiting the growth of *Toxoplasma* parasites [42,67,68]. To test the role of GRA57 in tryptophan-dependent parasite restriction, we supplemented the HFF growth media with L-tryptophan (L-Trp) as previously described in [44]. No rescue of the increased IFNγ susceptibility of RHΔGRA57 parasites was observed, suggesting GRA57 does not act through this pathway (S5 Fig). We observed no major rescue in parasite survival for the RHΔUPRT strain upon L-Trp supplementation, indicating that tryptophan metabolism is not the primary restriction mechanism in the HFFs used here. This is in line with other studies that have shown this pathway plays a minimal role in restriction of parasite growth in some HFFs, a discrepancy potentially dependent on the origin of the cells used [41,44].

We next assessed if GRA57 played a role in dense granule effector export into the host cell. Using RHΔMYR1 and RHΔMYR3 parasites as positive controls, we measured the ability of RHΔGRA57 parasites to induce host c-Myc nuclear translocation via IFA, which in wild-type parasites is induced through MYR-dependent export of GRA16 to the host nucleus [69]. Unlike the near ablation of c-Myc nuclear signal observed upon infection with RHΔMYR1 or RHΔMYR3 parasites, as previously published [15,16], at 24 h post-infection, there was no significant reduction in c-Myc nuclear signal with RHΔGRA57 parasites (S6 Fig). As GRA57 and GRA70 localise to the IVN and PVM, we next assessed the ultrastructure of RHΔGRA57 parasites via transmission electron microscopy (TEM), which showed that the IVN and PV form normally in this strain (S7 Fig). Together, this confirms that GRA57 is not directly important for MYR-dependent protein translocation or biogenesis of the parasite vacuole.

To comprehensively determine if GRA57 has any effect, directly or indirectly, on the host cell transcriptional response to *Toxoplasma*, we compared transcript levels using RNA-Seq of infected HFFs that were untreated or pretreated with IFNγ. Interferon-stimulated genes such as CXCL9, CXCL10, CXCL11, GBPs, and IDO1 were strongly up-regulated in HFFs upon IFNγ treatment, as previously reported [70]. Despite strong induction of GBPs in HFFs, GBPs are dispensable in HFF control of Toxoplasma [44], and therefore unlikely to mediate the increased clearance of GRA57 knockouts. In contrast to the effect of IFNγ, very few host genes were differentially expressed upon infection with RHΔGRA57 parasites compared to WT parasites (S7 Data). Of the host genes that were differentially expressed between parasite strain

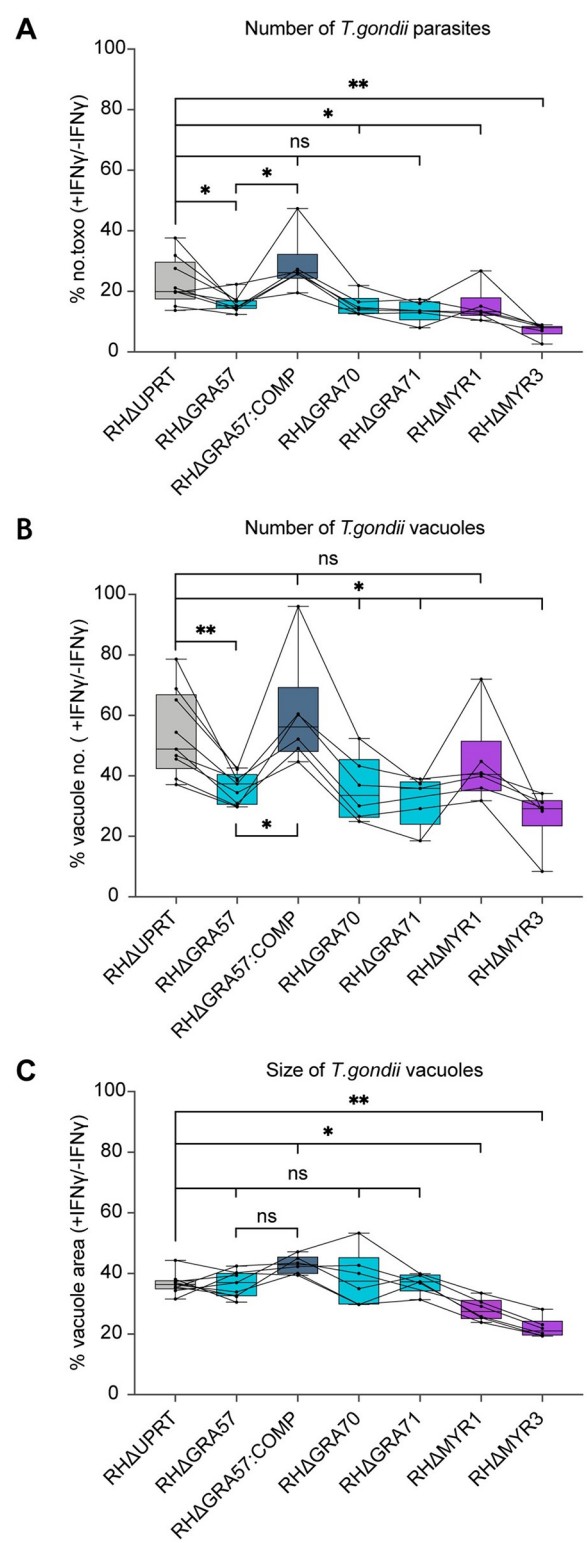

**Fig 4. GRA57/GRA70/GRA71 all contribute equally to resisting IFNγ-induced vacuole destruction in human fibroblasts. (A–C)** HFFs were pre-stimulated for 24 h with 100 U/ml IFNγ or left untreated then infected in technical triplicate with the indicated parasite strains for 24 h at an MOI of 0.3. Parasite survival was quantified through automated high-content imaging. (**A**) Parasite survival calculated as total number of *Toxoplasma* in IFNγ-stimulated cells as a percentage of the total in unstimulated cells. (**B**) Parasite survival calculated as number of vacuoles in IFNγ-

stimulated cells as a percentage of the number in unstimulated cells. (**C**) Parasite survival calculated as mean vacuole area per well [px$^2$] in IFNγ-stimulated cells as a percentage of the area in unstimulated cells. Data displayed as median survival with individual biological replicates overlaid. *p*-values were calculated by paired two-sided *t* test. *, *p* < 0.05; **, *p* < 0.01; ns, not significant. Source data can be found in S5 Data. HFF, human foreskin fibroblast.

infections, expression was not restored to WT infection levels upon infection with RHΔGRA57::GRA57-HA parasites. *Toxoplasma*-aligned transcripts revealed no significant differences in GRA70 or GRA71 expression levels in the RHΔGRA57 strain (S8 Data), indicating there is no compensatory overexpression of these genes following GRA57 knockout. Overall, there were very few changes in parasite gene expression between RHΔKu80 and RHΔGRA57, irrespective of IFNγ stimulation, indicating that deletion of GRA57 does not result in parasite transcriptional changes. Collectively, this data shows that GRA57 plays no role in protein export and does not function through significantly altering host or parasite gene expression.

## Deletion of GRA57/GRA70/GRA71 reduces ubiquitination on the parasite vacuole

Ubiquitination of the PVM is a marker for parasite clearance in IFNγ-activated human cells [40,48–50]. We therefore assessed if deletion of GRA57, GRA70, or GRA71 leads to differences in ubiquitination of the PVM. To test this, we measured total ubiquitin recruitment at 3 h post-infection. Recruitment was automatically scored by measuring the ubiquitin signal around each vacuole relative to the cytoplasmic background (Fig 5A).

Surprisingly, we found that cells infected with knockouts of GRA57 or its 2 interaction partners GRA70 and GRA71 had a significantly lower proportion of ubiquitin-recruited vacuoles in both unstimulated (S8A Fig) and IFNγ-pre-stimulated HFFs (Fig 5B). This reduction was also present for PRUΔGRA57 relative to PRUΔUPRT, indicating that this is a strain-independent effect. For RHΔGRA57, the reduction in ubiquitin targeting was restored to wild-type levels upon complementation. In contrast, RHΔMYR1 and RHΔMYR3 vacuoles were equally susceptible to ubiquitin recruitment as RHΔUPRT, providing further support that the GRA57 containing protein complex functions independently of the MYR translocon and its exported effectors.

Given that we observed only a minor increase in restriction in murine fibroblasts upon GRA57 deletion, we next investigated the ubiquitin recruitment levels to these strains in MEFs. Recruitment levels were much higher in PRU than RH, a phenotype which has previously been shown to be dependent on the strain-specific expression of GRA15 [41]. Despite overall lower levels of ubiquitin recruitment, deletion of GRA57, GRA70, or GRA71 in the RH strain led to a similar relative reduction in ubiquitin-targeting in IFNγ-activated MEFs as in HFFs of approximately 50% (Fig 5C). This reduction was again rescued for RHΔGRA57 upon complementation. PRUΔGRA57 displayed a smaller but still significant reduction in ubiquitination relative to wild-type in comparison to HFFs. In unstimulated MEFs, the percentage of ubiquitin-positive vacuoles was lower than in unstimulated HFFs; however, deletion of GRA57, GRA70, or GRA71 still resulted in a trend towards reduced ubiquitin recruitment (S8B Fig). Together, this data suggests that as RHΔGRA57 parasites are much more sensitive to IFNγ-mediated clearance in HFFs than in MEFs, this decrease in ubiquitin recruitment does not correlate with increased susceptibility to IFNγ-induced clearance and may therefore not be directly linked to the parasite clearance mechanism.

The type of ubiquitin linkage recruited to the PV can dictate the fate of *Toxoplasma* vacuoles [40,41,49], so we next tested if the reduction in ubiquitin recruitment was a global or

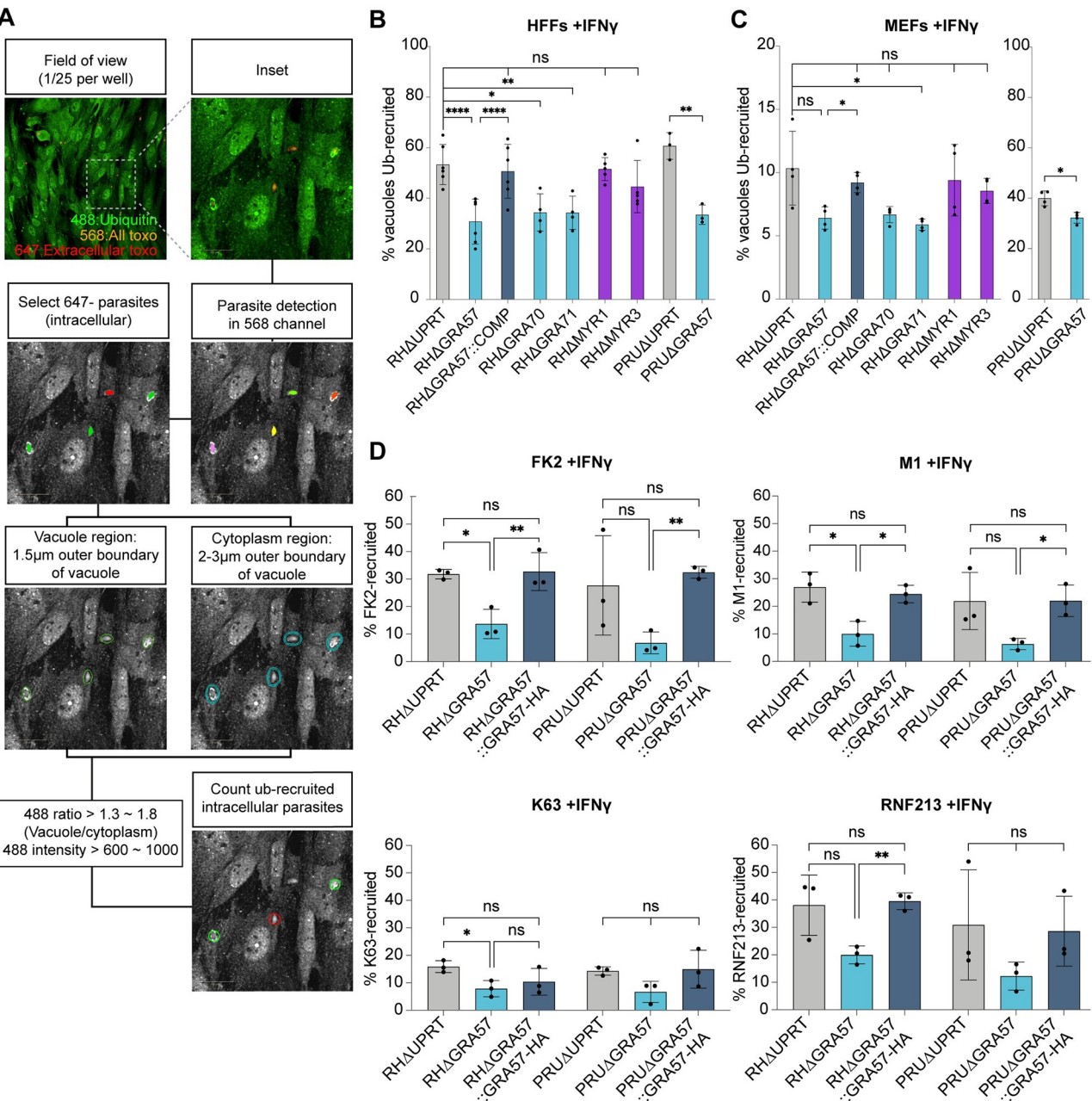

**Fig 5. Deletion of GRA57, GRA70, or GRA71 leads to reduced host ubiquitin recruitment to the PVM. (A)** Schematic of automated high-content imaging analysis pipeline to determine ubiquitin recruitment levels. **(B, C)** Recruitment of total host ubiquitin (FK2) to *Toxoplasma* vacuoles in **(B)** HFFs and **(C)** MEFs. Host cells were pre-stimulated with 100 U/ml IFNγ for 24 h, infected with indicated lines for 3 h prior to fixation and staining for total ubiquitin. Recruitment of ubiquitin was automatically counted using high-content imaging and analysis. **(D)** Recruitment of total ubiquitin (FK2), K63-linked ubiquitin, linear ubiquitin (M1), and the E3 ligase RNF213 to *Toxoplasma* vacuoles. HFFs were pre-stimulated with 100 U/ml IFNγ for 24 h, infected with indicated lines for 3 h prior to fixation and staining. Recruitment was automatically quantified for FK2, M1, and RNF213. K63 recruitment was manually scored, with minimum 100 vacuoles scored per condition. *p*-values were calculated by paired two-sided *t* test. *, $p < 0.05$; **, $p < 0.01$; ***, $p < 0.001$; ****, $p < 0.0001$; ns, not significant. Source data can be found in S9 Data. HFF, human foreskin fibroblast; MEF, mouse embryonic fibroblast; PVM, parasitophorous vacuole membrane.

linkage-specific effect. We found that M1- and K63-linked ubiquitin recruitment was reduced on ΔGRA57 parasite vacuoles relative to wild-type, in unstimulated (S8C Fig) and IFNγ-pre-stimulated HFFs (Fig 5D). M1 and K63 linkages were significantly reduced on RHΔGRA57 vacuoles, with a similar but not statistically significant trend observed for PRUΔGRA57 vacuoles (Fig 5D). K48-linked ubiquitin was found at very low levels across all conditions tested (S8D Fig), in line with previous observations [40,41,49]. As M1- and K63-linked ubiquitination of *Toxoplasma* vacuoles has recently been shown to be dependent on the host E3 ligase RNF213 in HFFs [49,50], we next tested if RNF213 was recruited to ΔGRA57 vacuoles. RNF213 recruitment was also reduced at ΔGRA57 vacuoles in unstimulated (S8C Fig) and IFNγ-stimulated cells (Fig 5D). Together, this suggests that the observed reduction in ubiquitination of ΔGRA57 vacuoles could be a secondary consequence of reduced vacuole recognition by RNF213.

Recent work has shown that parasites are cleared in HFFs through endolysosomal acidification of the PV and subsequent killing of the parasites [41]. We hypothesised that RHΔGRA57 vacuoles could be initially ubiquitinated to the same level as wild-type parasites but cleared by the host cell prior to fixation and staining at 3 h. To exclude this possibility, we performed the same recruitment analysis in HFFs at 1 and 2 h post-infection to determine if ubiquitin recruitment to RHΔGRA57 vacuoles peaked at an earlier time point. This showed that RHΔGRA57 vacuoles are consistently less targeted by ubiquitin than RHΔUPRT at these early stages post-infection, in both the untreated and pre-stimulated HFFs (S9 Fig). These results do not exclude the possibility that RHΔGRA57 vacuoles are cleared more rapidly during the first hour of infection; however, we think this is unlikely given that RHΔGRA57 vacuoles are also less susceptible to ubiquitination in unstimulated HFFs, yet display no growth phenotype in this context (S1C Fig). Together, this data confirmed that RHΔGRA57 vacuoles are continuously less susceptible to ubiquitin targeting following host cell invasion.

While this manuscript was in review, another group showed that deletion of GRA57, GRA70, or GRA71 promotes early parasite egress in IFNγ-stimulated HFFs [71]. We therefore hypothesised that the reduction in ubiquitination could be a result of early egress of ubiquitin-recruited parasites. To test this hypothesis, we performed the same recruitment analysis in HFFs with the addition of Compound 2, an inhibitor of the cGMP-dependent protein kinase G (PKG) that is essential for parasite egress [72]. Compound 2 was added at 30 min post-infection to inhibit parasite egress occurring prior to fixation at 3 h post-infection. There was no increased ubiquitination of ΔGRA57, ΔGRA70, or ΔGRA71 vacuoles upon addition of Compound 2 (S10A Fig), indicating that early egress is not responsible for the reduced ubiquitin recruitment. Additionally, this data showed that while there was mild IFNγ-dependent restriction of both RH and PRU parasites at this early time point, there was no significant difference in survival between any of the strains tested or upon addition of Compound 2 (S10B Fig). Together, this suggests that IFNγ-induced early egress of *Toxoplasma*, and the enhanced clearance of ΔGRA57, ΔGRA70, and ΔGRA71 parasites, occurs later than 3 h post-infection of HFFs.

## Discussion

In this study, we used targeted CRISPR screens to identify *Toxoplasma*-secreted virulence factors that protect the parasite from human cell-autonomous immunity. CRISPR screens were carried out in RH and PRU *Toxoplasma* strains, revealing a novel complex of 3 interacting dense granule effectors, comprising GRA57, GRA70, and GRA71, that contribute equally to *Toxoplasma* resistance to IFNγ-induced responses in HFFs.

In addition to GRA57 and its 2 partner proteins, we show that 2 components of the MYR effector export machinery, MYR1 and MYR3, are important for parasite survival in IFNγ-

activated HFFs in both pooled and single knockout infections. Other MYR components including the kinase ROP17 [17] displayed an IFNγ-survival phenotype in both screens, whereas the effector chaperone GRA45 [55] was essential for IFNγ survival in only the PRU screen. Together, this would suggest that the GRA protein translocation machinery of *Toxoplasma* is required for survival in IFNγ-treated HFFs. However, MYR2 and MYR4 displayed no phenotype in either of our screens, though it is possible these are false-negative results. Only MYR1 and MYR3 have previously been shown to stably associate with each other *in vitro* [16], therefore, whether MYR1 and MYR3 have an important additional function in HFFs that is distinct from effector export would be interesting to explore in the future.

In contrast, neither ROP18 nor GRA12 had a major phenotype in IFNγ-stimulated HFFs, whereas in murine screens ROP18 and GRA12 consistently emerge as the most important secreted effectors for parasite survival [22,54,55]. This emphasises the importance of using human *in vitro* systems to identify parasite effectors that mediate the pathogenesis of *Toxoplasma* in the human host. In both the RH and PRU screens, we observed high levels of parasite restriction in HFFs, with approximately 90% of the mutant population restricted in cells pre-stimulated with 5 U/ml IFNγ, a dose 20-fold less than that commonly used in literature. HFFs are extensively used for the continuous culture of *Toxoplasma* and are often considered as "non-immune" cells, but these results support that structural cells can be important components of the innate immune response to *Toxoplasma* infection [73]. Other human non-hematopoietic cells have also been demonstrated to restrict *Toxoplasma* in response to IFNγ signalling, including neurons [74], epithelial [39] and endothelial cells [40].

High-content imaging of parasite restriction in HFFs revealed the RHΔMYR1 and RHΔMYR3 strains are highly susceptible to restriction through both increased vacuole clearance and growth restriction. Given the pleiotropic effects on the host that are mediated by MYR-dependent effector export [21], we assume that the abrogation of all effector export leads to hypersensitivity to multiple IFNγ-induced restriction pathways in HFFs. This is further supported as no other dense granule protein with an equally strong phenotype was identified in our screen, although it is possible that targets were filtered out due to variability between guides. As our library is constructed from hyperLOPIT prediction [12], it could also be possible that some GRAs or ROPs are omitted if not previously annotated as localised to the dense granules or rhoptries. In contrast to MYR1 and MYR3 knockouts, single knockouts of GRA57, GRA70, or GRA71 were more sensitive to IFNγ-mediated vacuole clearance but did not display decreased replication in activated cells. As GRA57 was not shown to influence effector export, and both GRA57 and GRA70 do not appear to localise outside of the PV, we assume that this dense granule complex is not exported via the MYR translocon and functions independently of the MYR complex to promote parasite survival in IFNγ-activated cells.

In contrast to HFFs, we observed no significant impact on IFNγ survival in MEFs upon deletion of GRA57, with no previously identified phenotype for GRA57/GRA70/GRA71 in murine screens [22,54,55,64]. From this, we infer that the trimeric complex functions to subvert an IFNγ-induced response that is present in HFFs but absent or less dominant in MEFs. One such response is the induction of HFF cell death and concomitant premature parasite egress [44]. While this manuscript was under review, another group found a role for GRA57, GRA70, and GRA71 in preventing premature egress in IFNγ-stimulated HFFs [71]. We observed a larger IFNγ-dependent decrease in vacuole numbers when HFFs were infected with knockouts of the GRA57/GRA70/GRA71 complex, an effect which could be explained by early egress of these strains in response to IFNγ. The mechanism through which IFNγ induces early egress however remains unknown.

It has previously been shown that endolysosomal acidification of the PV mediates clearance of PRU parasites in IFNγ-activated HFFs, an effect that is inhibited through unknown

mechanisms in RH parasites [41]. Acidification of the PV during the normal lytic cycle of *Toxoplasma* serves as a signal for microneme secretion and parasite egress [75]. Therefore, acidification induced by host cell endolysosomal fusion with the PV may induce premature parasite egress, thereby limiting parasite replication and dissemination. Future work will aim to determine if the GRA57/GRA70/GRA71 complex functions to resist IFNγ-induced vacuole acidification in HFFs.

Autophagy has emerged as a major clearance mechanism for *Toxoplasma* [76] and multiple other intracellular pathogens [77]. In our efforts to determine if host autophagy contributed to the increased clearance of GRA57/GRA70/GRA71 knockouts, we unexpectedly found a marked reduction in the percent of these knockout vacuoles targeted by host ubiquitin in HFFs and MEFs, in both the presence and absence of IFNγ-stimulation. For GRA57 knockouts, we determined there was specific depletion of K63- and M1-linked ubiquitin chains at the PVM, correlating with reduced recruitment of the host E3 ligase RNF213, which has been demonstrated to facilitate attachment of these ubiquitin linkages to the PVM [49]. We believe there is unlikely to be a causal link between ubiquitin recruitment and the increased restriction of GRA57/GRA70/GRA71 knockouts in IFNγ-activated HFFs, given that (a) reduced ubiquitin recruitment was also observed in MEFs, in which the knockouts strains are not as susceptible to IFNγ; (b) reduced RNF213 and ubiquitin recruitment was also evident in unstimulated cells; and (c) a decrease in vacuole ubiquitination should correlate with an increase in parasite survival.

Mukhopadhyay and colleagues [41] have shown that ubiquitination of *Toxoplasma* vacuoles in HFFs is not intrinsically linked to downstream fusion with the endolysosomal system, supporting that our observed decrease in vacuole ubiquitination may be a secondary effect of deletion of the GRA57/GRA70/GRA71 complex. At 3 h post-infection, the total percentage of vacuoles with ubiquitin recruitment is reduced in these knockouts, but not entirely ablated (Fig 5B and 5D). Therefore, an alternative possibility is that in the absence of the complex, the vacuoles become hypersensitive to the remaining ubiquitination and subsequent disruption.

Recent work has found that RNF213, an IFNγ-induced host E3 ubiquitin ligase, mediates the majority of ubiquitin recruitment to both RH and PRU *Toxoplasma* vacuoles in HFFs [49,50]; however, the target of this ligase is currently unknown. The observation that GRA57 deletion in both RH and PRU *Toxoplasma* leads to reduced recruitment of RNF213 to the PV poses the intriguing question of whether the GRA57/GRA70/GRA71 complex is required for RNF213 recruitment, is a direct target of ubiquitination, or is responsible for correct positioning of an RNF213 target at the PV. Each of these possibilities will be interesting to follow up to better understand the function of RNF213 in human cell-autonomous restriction of *Toxoplasma*.

In conclusion, we identify several proteins that are required for survival in HFFs under conditions of IFNγ restriction in 2 *Toxoplasma* strains that differ in virulence. We identify a complex of 3 proteins that is required to protect *Toxoplasma* parasites from killing by the host cell. While deletion of the complex components leads to reduced ubiquitination of the parasite vacuole, we suggest that functional consequences of the complex deletion likely go beyond a role in ubiquitination.

Our study did not identify which host cell pathway is counteracted by the GRA57/GRA70/GRA71 complex, though we ruled out a role in effector export, transcriptional modulation, tryptophan metabolism, and vacuole ultrastructure formation. Data described by Krishnamurthy and colleagues [71] suggests a role for the GRA57/GRA70/GRA71 complex in preventing IFNγ-induced premature egress, but the host factors mediating this phenotype remain to be identified. Future work to gain mechanistic insight into this complex would ideally use an unbiased knockout or knockdown screen of host IFNγ-stimulated genes, similar to that

described by Matta and colleagues [50], to identify which host pathways are counteracted by the GRA57/GRA70/GRA71 complex. This would also provide a basis for further work aimed at understanding the biochemical functions mediated by the complex.

It is important to note that none of the 3 proteins identified here to protect the parasite against the IFNγ response in human cells have been identified in previous screens in murine cells or mice. Given this discrepancy, it will be interesting to determine if this complex targets a host pathway not present or active in murine cells. As humans are primarily an accidental host for *Toxoplasma*, it is highly likely that the GRA57 complex evolved to protect the parasite in species other than humans. Comparison of GRA57 and MYR function in cells of various origins may therefore reveal commonalities in the cell autonomous immune response between humans and other species that can be infected by *Toxoplasma*.

## Methods

### CRISPR-Cas9 screens

**Parasite transfections.**   To generate a pooled population of parasite effector knockouts, we used a plasmid library comprising 1,299 single-guide RNAs (sgRNAs) targeting 253 predicted secreted proteins, with an average of 5 sgRNAs/gene, with sgRNAs integrated into a pCas9-GFP-T2A-HXGPRT::sgRNA vector. The plasmid pool was linearised overnight with *Kpn*I-HF (NEB, R3142), purified by phenol-chloroform precipitation, then resuspended in P3 (Lonza, V4XP-3024) transfection buffer at a concentration of 1 µg/µl.

A minimum of 90 million parasites (RHΔHXGPRT or PRUΔHXGPRT) were transfected in triplicate using an Amaxa 4D Nucleofector (Lonza, AAF-1003X) with pulse code EO-115, with 30 µg/transfection of the purified sgRNA plasmid library. Transfected populations were selected for after 24 h using 25 µg/mL mycophenolic acid (Sigma–Aldrich, 89287) and 50 µg/mL xanthine (Sigma–Aldrich, X3627) (M/X). Transfection efficiency was assessed via plaque assay, achieving a final guide coverage of 174X and 68X in the RH and PRU screens, respectively. Three days post-transfection, parasites were syringe lysed and added to HFF monolayers with 100 U/ml benzonase (Merck, E1014-25KU) overnight to remove traces of input DNA. Eight days post-transfection, subset parasite populations were harvested for genomic DNA (gDNA) extraction to determine guide abundance in the starting inoculum. Remaining triplicate transfections were combined to generate the inoculum used for infections.

**Infections.**   HFFs were grown to confluency in T175 flasks, then stimulated with 5 U/ml human IFNγ (Bio-Techne, 285-IF-100) for 24 h pre-infection. For infections, parasites were isolated from HFFs by syringe lysis through a 30-gauge needle (×3) and passed through 5 µm filters (Millipore, SMWP04700), then added to HFFs at an MOI of 0.2 ($1.4 \times 10^6$ parasites/flask) for 48 h. At 48 h post-infection, parasites were harvested as above from HFFs, counted, and then a subset were added to new flasks of IFNγ treatment-matched HFFs at an MOI of 0.2 for a further 48 h. For each round of infection surviving parasites populations were expanded in unstimulated HFFs for a further 48 h prior to harvesting for gDNA extraction.

**gRNA isolation and sequencing.**   Genomic DNA was extracted from samples using Qiagen DNEasy Blood kit, then guide sequences were amplified from gDNA and the plasmid pool by nested PCR using KAPA HIFI Hotstart PCR kit (Kapa Biosystems, KK2501), as previously described in [22]. Primers used for nested PCR are listed in S11 Data (primers 1–20). Purified PCR products were then sequenced on a HiSeq400 (Illumina) with single end 100 bp reads at a minimum read depth of 5 million reads/sample.

**Sequencing data analysis.**   gRNA sequences were aligned to a reference library as described previously in [22,54]. The lowest 1.5 percentile of guides expressed across all samples were removed from the analysis. Counts were normalised using the median of ratios, then

genes represented by fewer than 3 matching guides were removed from the analysis. The median L2FC for each gene was calculated from the normalised counts at the end of growth in pre-stimulated HFFs relative to unstimulated HFFs, or from counts at the end of growth in unstimulated HFFs relative to the counts in the starting inoculum sample. The median absolute deviation (MAD) score across gRNA L2FCs targeting each gene was calculated, and genes with the highest 1.5% of MAD scores were removed from the analysis. A DISCO score based on the local FDR-corrected q-value was calculated for each L2FC to compare between untreated and IFNγ pre-stimulated HFFs.

## Parasite and host cell culture

*Toxoplasma gondii* strains were maintained by serial passage every 2 to 3 days in HFFs (ATCC, SCRC-1041). For experiments, parasites were isolated by syringe lysis through a 27-gauge needle and passed through 5 μm filters. Parasite genotypes were verified by restriction fragment length polymorphism analysis of the SAG3 gene using primers 70 and 71 [78]. HFFs and MEFs (gift from Felix Randow) were maintained in DMEM with GlutaMAX (Gibco) supplemented with 10% foetal bovine serum (Gibco). Parental strains used were RHΔHXGPRT and PRUΔHXGPRT [79], RHΔKU80 [80], and PRUΔKU80 [81].

## Generation of parasite cell lines

**Knockouts.** For generation of RHΔGRA57 and PRUΔGRA57 strains, 2 guides were designed targeting exon 1 and exon 7 of the GRA57 coding sequence (CDS). Guides were integrated separately into a pCas9-GFP::sgRNA vector by inverse PCR using a general reverse primer [21] and primers 24 or 25. The exon 1 targeting plasmid was digested with XhoI/KpnI (NEB), and the exon 7 guide was amplified with primers 22 and 23 to facilitate Gibson cloning of the exon 7 guide into the same backbone. For all other knockouts generated in this study (RHΔMYR3, RHΔGRA70, and RHΔGRA71), single-guide plasmids were generated using primers 26–28. Homology repair cassettes were generated by PCR amplification from Pro-$^{GRA1}$-mCherry-T2A-HXGPRT-Ter$^{GRA2}$, using primers with 40 bp flanking homology to the 5′ and 3′ UTRs of each gene (primers 29–36). A total of 10 μg of repair cassette was purified with 5 μg of pCas9-GFP::sgRNA and transfected into RHΔKu80 or PRUΔKu80 parasites, using an Amaxa 4D Nucleofector as described above. Transfectants were selected 24 h post-transfection with 25 μg/ml of mycophenolic acid (Merck) and 50 μg/ml xanthine (Sigma) (M/X) and cloned by limiting dilution. Integration of the mCherry-HXGPRT repair templates was verified by PCR using primers 37–46.

**Complementation.** To complement the RHΔGRA57 and PRUΔGRA57 strains, the 5′ UTR and CDS of GRA57 was generated through a combination of PCR amplification from gDNA (5′ UTR, exons 1 and 7- primers 47–54) and amplification from gBlocks (IDT) for exons 2–4 and exon 6 (listed in S11 Data). PCR products were inserted into the pUPRT vector [82] by Gibson assembly. The pUPRT:GRA57-HA plasmid was linearised with ScaI, then 10 μg was transfected into RHΔGRA57 or PRUΔGRA57 alongside 1 μg of pCas9-GFP::UPRT. Transfectants were selected 24 h post-transfection with 5 μM 5′-fluo-2′-deoxyuridine (FUDR), then cloned and verified by PCR at the UPRT locus using primers 55–58.

**Endogenous tagging.** To endogenously tag GRA57 with an HA epitope tag at the C-terminus, a pCas9-GFP::sgRNA plasmid targeting the 3′ end of exon 7 of GRA57 was generated by inverse PCR from pCas9-GFP::sgRNA using primers 21 and 59. A repair cassette with 40 bp flanking homology was amplified by PCR from HA-Ter$^{GRA2}$::Pro$^{DHFR}$-HXGPRT-Ter$^{DHFR}$ using primers 61 and 62. Approximately 10 μg of repair cassette was purified with 5 μg of pCas9-GFP::sgRNA and transfected into RHΔKu80 as above. For endogenous tagging of

GRA70 with a C-terminal V5 tag, the pCas9-GFP::sgRNA plasmid was generated using primers 21 and 60, and the repair cassette was generated by PCR from a V5-Ter$^{GRA2}$::Pro$^{DHFR}$-HXGPRT-Ter$^{DHFR}$ construct, using primers 63 and 64. Transfectants were selected 24 h post-transfection with 25 μg/ml of mycophenolic acid (Merck) and 50 μg/ml xanthine (Sigma) (M/X), then cloned and verified by PCR using primers 65–69.

All primers used for generating parasite lines are listed in S11 Data.

## Plaque assays

A total of 100 parasites were added to HFF monolayers in a T25 flask and allowed to grow undisturbed for 10 days. Cells were fixed with 100% methanol and stained with crystal violet, then plaques were imaged on a ChemiDoc imaging system (BioRad). Plaque area was measured in FIJI [83]. Differences between strains were determined by one-way analysis of variance (ANOVA) with Tukey's multiple comparison test.

## Co-immunoprecipitation mass-spectrometry

**Immunoprecipitation.** HFFs grown to confluency in T175 flasks were pre-stimulated with 2.5 U/ml IFNγ (Bio-Techne, 285-IF-100) for 6 h prior to infection with RHΔKu80 or RHGRA57-HA in triplicate, and 24 h post-infection, infected cells were washed 3× in cold PBS then lysed in cold immunoprecipitation (IP) buffer (10 mM Tris, 150 mM NaCL, 0.5 mM EDTA + 0.4% NP40, pH 7.5 in H$_2$O, supplemented with 2× cOmplete Mini EDTA-free Protease Inhibitor Cocktail). Lysates were syringe lysed 6× through a 30 g needle, then centrifuged at 2,000 g for 20 min to remove the insoluble fraction. Soluble fractions were added to 50 μl/sample anti-HA agarose matrix (Thermo), then incubated overnight at 4˚C with rotation. The matrix was washed 3 times with IP buffer, then proteins were eluted in 30 μl 3× Sample Loading Buffer (NEB) at room temperature for 10 min.

**Mass spectrometry.** Approximately 20 μl of each IP elution was loaded on a 10% Bis-Tris gel and run into the gel for 1 cm, then stained with InstantBlue Coomassie Protein Stain. Proteins were alkylated in-gel prior to digestion with 100 ng trypsin (modified sequencing grade, Promega) overnight at 37˚C. Supernatants were dried in a vacuum centrifuge and resuspended in 0.1% trifluoroacetic acid (TFA), and 1 to 10 μl of acidified protein digest was loaded onto a 20 mm × 75 μm Pepmap C18 trap column (Thermo Scientific) on an Ultimate 3000 nanoRSLC HPLC (Thermo Scientific) prior to elution via a 50 cm × 75 μm EasySpray C18 column into a Lumos Tribrid Orbitrap mass spectrometer (Thermo Scientific). A 70' gradient of 6% to 40% B was used to elute bound peptides followed by washing and re-equilibration (A = 0.1% formic acid, 5% DMSO; B = 80% ACN, 5% DMSO, 0.1% formic acid). The Orbitrap was operated in "Data Dependent Acquisition" mode followed by MS/MS in "TopS" mode using the vendor supplied "universal method" with default parameters.

Raw files were processed to identify tryptic peptides using Maxquant (maxquant.org) and searched against the *Toxoplasma* (ToxoDB-56_TgondiiGT1_AnnotatedProteins) and Human (Uniprot, UP000005640) reference proteome databases and a common contaminants database. A decoy database of reversed sequences was used to filter false positives, at peptide and protein false detection rates (FDRs) of 1%. *T* test-based volcano plots of fold changes were generated in Perseus (maxquant.net/perseus) with significantly different changes in protein abundance determined by a permutation-based FDR of 0.05% to address multiple hypothesis testing.

The mass spectrometry proteomics data have been deposited to the ProteomeXchange Consortium via the PRIDE partner repository with the dataset identifier PXD041352.

## Protein secondary structure prediction

Protein sequences for GRA57, GRA70, and GRA71 from ToxoDB were used to predict boundaries and regions of secondary structure with PSIPRED [84]. Transmembrane domains (TMDs) were assigned using MEMSAT-SVM (within PSIPRED), DeepTMHMM [85], and CCTOP [86], with TMDs only annotated where ≥2 programmes concurred. Regions of predicted coiled-coil were determined using Waggawagga [87], with coiled-coil annotated where the probability score was ≥90.

## Western blotting

Parasites were isolated from host cells by syringe-lysis through 27-gauge needles, passed through 5 μm filters and washed ×1 with cold PBS. Samples were lysed on ice in 1% NP40 IP buffer (10 mM Tris, 150 mM NaCL, 0.5 mM EDTA + 1% NP40, pH 7.5 in $H_2O$, supplemented with 1× cOmplete Mini EDTA-free Protease Inhibitor Cocktail) for 30 min, then spun at 2,000 g for 15 min to remove the insoluble fraction. Approximately 10 μg of protein per sample was incubated with 3× loading buffer for 10 min at room temperature, then separated by SDS-PAGE on a NuPAGE 3% to 8%, Tris-Acetate gel. Proteins were transferred to a nitrocellulose membrane using the High Molecular Weight protocol on the Trans-Blot Turbo transfer system (Bio-Rad). Membranes were blocked in 5% milk in 0.05% Tween 20 in PBS for 1 h at room temperature, followed by incubation for 1 h at room temperature with primary antibodies diluted in 1% milk in 0.05% Tween 20 in PBS. Blots were then incubated with appropriate secondary antibodies for 1 h at room temperature. Primary antibodies used were 1:200 mouse anti-*T. gondii* (Santa Cruz, SC-52255), 1:1,000 rat anti-HA-Peroxidase (Roche, 12013819001), and 1:1,000 rabbit anti-V5 (Abcam, AB_2809347). Secondary antibodies used were 1:10,000 goat anti-mouse HRP (Insight Biotechnologies, 474–1806) and 1:3,000 goat anti-rabbit HRP (Insight Biotechnology, 474–1506). HRP was detected using an enhanced chemiluminescence (ECL) kit (Pierce), visualised on a Chemi-Doc imaging system (BioRad).

## Transmission electron microscopy

HFFs were grown to confluency on 13-mm glass coverslips. HFFs were infected with 60,000 parasites per strain for 24 h, then washed 1× with DPBS prior to fixation in 2.5% glutaraldehyde + 4% formaldehyde in 0.1 M phosphate buffer (PB) for 30 min. Samples were washed 2× with 0.1 M PB and stained with 1% (v/v) osmium tetroxide (Taab)/1.5% (v/v) potassium ferricyanide (Sigma) for 1 h at 4˚C. Samples were washed 2× in dH2O and were then transferred to a Pelco BioWave Pro+ microwave (Ted Pella, Redding, United States of America) for a further 2 dH2O washes in the Biowave without vacuum (at 250 W for 40 s). The SteadyTemp plate was set to 21˚C. In brief, the samples were incubated in 1% (w/v) tannic acid in 0.05 M PB (pH 7.4) (Sigma) for 14 min under vacuum in 2-min cycles alternating with/without 100 W power, followed by 1% sodium sulphate in 0.05 M PB (pH 7.4) (Sigma) for 1 min without vacuum at 100 W. The samples were then washed in dH2O and dehydrated in a graded ethanol series (25%, 50%, 70%, 90%, and 100%, twice each) and in acetone (3 times) at 250 W for 40 s without vacuum. Exchange into Epon 812 resin (Taab) was performed in 25%, 50%, and 75% resin in acetone, at 250 W for 3 min, with vacuum cycling (on/off at 30-s intervals). The samples were transferred to 100% resin overnight before embedding at 60˚C for 48 h. A total of 70 nm sections were sliced with a diamond knife on an RMC Powertome Ultramicrotome and sections analysed by a JEM-1400 FLASH transmission electron microscope (Jeol) with Jeol Matataki Flash sCMOS camera.

## Immunofluorescence assays

HFFs were seeded to confluency on 15-mm glass coverslips and infected with 60,000 parasites/well for 24 h. For images shown in S4 Fig, HFFs were pre-stimulated with 100 U/ml IFNγ for 24 h prior to infection. Infected cells were fixed in 4% paraformaldehyde (PFA) for 15 min, then permeabilised in 0.2% Triton-X100 (GRA2 staining) or in 0.1% saponin (GRA3 staining) for 10 min and blocked with 3% bovine serum albumin (BSA) in DPBS for 30 min. Primary and secondary antibodies used for immunofluorescence assays are listed in S13 Data. All antibodies were incubated in 3% BSA in DPBS for 1 h at room temperature, with 3× washes in DPBS between stages. Final secondary staining was combined with DAPI (5 μg/ml). Slides were mounted in ProLong Gold (Thermo Fisher, P36930). Images were acquired on a Visi-Tech instant SIM (VT-iSIM) microscope using a 150× oil-immersion objective with 1.5 μm z axis steps. Resultant stacks were deconvoluted and processed using Microvolution plugin in FIJI [83].

## c-Myc nuclear translocation assays

HFFs were grown to confluence in 8-well ibidi μ-slides (Ibidi, 80806). Cells were infected with 30,000 parasites in DMEM with 0.1% FBS. After 24 h, slides were fixed in 4% PFA for 15 min, permeabilised in 0.2% Triton-X100 for 10 min, then blocked in 3% BSA for 30 min. Host c-Myc was stained using 1:800 rabbit anti-cMyc for 2 h at room temperature, followed by 1:1,000 anti-Rabbit-AlexaFluor488 and 5 μg/ml DAPI for 1 h at room temperature. Images were acquired on a Nikon Ti-E inverted widefield fluorescence microscope with a Nikon Plan APO 40×/0.95 objective and Hamamatsu C11440 ORCA Flash 4.0 camera. Host nuclear c-Myc signal of infected cells was measured in FIJI, and the median background c-Myc signal was subtracted for each image. A minimum of 100 infected cells were analysed per strain for each biological replicate. Data is shown as the median fluorescence intensity for each strain relative to RHΔUPRT in each biological replicate. Differences between strains were determined by one-way analysis of variance (ANOVA) with Tukey's multiple comparison test.

## Cytation plate reader assays

**Survival assays.** Host cells were seeded in 96-well black imaging plates (Falcon) to confluency, then media was changed to phenol red free DMEM (Gibco, A1443001) with or without 100 U/ml IFNγ (Human:Bio-Techne, 285-IF-100| Mouse: Thermo Fisher, Gibco PMC4031) for 24 h. Parasite strains were syringe lysed and added to host cells at 40,000 parasites/well. Plates were imaged live at 24 h post-infection (MEFs) or 48 h post-infection (HFFs) on a Cytation 5 plate reader using the 4× objective (HFFs) or 20× (MEFs). Total area with signal in the Texas Red channel (Ex/Em 586/647) was measured per well. Differences between strains were determined by paired two-sided *t* test.

**Tryptophan supplementation.** HFF were seeded as described above, then treated wells were supplemented with 1 mM L-tryptophan (VWR, J62508-09) dissolved in 0.1 N NaOH at the same time as IFNγ stimulation, as described in [44]. Untreated wells had 0.1 N NaOH added as a vehicle control. Data acquisition on a Cytation 5 plate reader was performed as with survival assays.

## High-content imaging survival assays

Host cells were seeded to confluency in 96-well imaging plates (Ibidi). Cells were pre-stimulated for 24 h with 100 U/ml IFNγ according to host species. Cells were infected at an MOI of 0.3 for 24 h, then fixed in 4% PFA and stained with 5 μg/ml DAPI and 5 μg/ml CellMask Deep Red (Invitrogen). Plates were imaged using the Opera Phenix high-content screening system, with 25 images and 5 focal planes acquired per well. Automated analysis of infection phenotypes was performed using Harmony v5 (PerkinElmer) as described in [54]. Data is reported as the mean proportion of each factor (total *Toxoplasma* number, vacuole size or vacuole number) in IFNγ-treated wells relative to untreated wells. Differences between strains were determined by paired two-sided *t* test.

## Vacuole recruitment assays

**Infections and staining.** Host cells were seeded and pre-stimulated as above in 96-well imaging plates (Ibidi). Cells were infected with 80,000 parasites/well and centrifuged at 300 g for 5 min to synchronise infection. At 3 h post-infection, cells were washed 3× in DPBS to remove extracellular parasites, then fixed in 4% PFA for 15 min and blocked with 3% BSA in DPBS for 1 h at room temperature. Extracellular parasites were stained prior to permeabilisation using 1:1,000 rabbit anti-toxo (Abcam, ab138698) and 1:500 goat anti-rabbit-AlexaFluor647 (Life Technologies, A21244). Cells were then permeabilised with 0.2% Triton-X100 for 10 min and re-blocked with 3% BSA. Host marker recruitment was probed for 2 h at room temperature using the following antibodies: mouse anti-total ubiquitin (1:200, Merck, ST1200), rabbit anti-K63 ubiquitin (1:100, Merck, 05–1308), rabbit anti-K48 ubiquitin (1:500, Sigma, ZRB2150), rabbit anti-M1 linear ubiquitin (1:200, Sigma, ZRB2114), or rabbit anti-RNF213 (1:1,000, Sigma, Human Protein Atlas no. HPA003347), followed by donkey anti-mouse-AlexaFluor488 (Thermo, A32766) or donkey anti-rabbit-AlexFluor488 (Thermo, A32790) for 1 h at room temperature. For measurement of K63 recruitment, cells were imaged on a Nikon Ti-E inverted widefield fluorescence microscope with a Nikon Plan APO 40×/0.95 objective, with at least 100 intracellular vacuoles scored/condition.

**Automated image acquisition and analysis.** Plates were imaged using the Opera Phenix high-content screening system, with 25 images and 5 focal planes from −1 to 1 μm with a step size of 0.5 μm acquired per well. Automated analysis was performed using Harmony v5 (PerkinElmer). Recruitment was automatically quantified by first excluding extracellular parasites, then measuring AF488 signal within the vacuole region (1.5 μm boundary of vacuole) and the cytoplasmic region around the vacuole (2 to 3 μm boundary of vacuole). Parasites with AF488 signal higher within this radius relative to the cytoplasmic background of each infected cell were classed as ubiquitin-recruited. Image acquisition parameters and analysis sequences with antibody-specific thresholds are detailed further in S12 Data. For each well, the percentage of ubiquitin-recruited intracellular vacuoles was determined, and then the mean percentage recruitment was calculated across triplicate wells. A median of 1,900 intracellular vacuoles were quantified per strain for each biological replicate, with a minimum number of 200. Differences between strains were determined by paired two-sided *t* test.

## Compound 2 inhibition of egress

Host cells were seeded and pre-stimulated as above in 96-well imaging plates (Ibidi). Cells were infected with 400,000 parasites/well and centrifuged at 300 g for 5 min to synchronise infection. At 30 min post-infection, cells were washed 3× in DPBS to remove uninvaded parasites, and then 5 μm Compound 2 (gift from Michael Blackman) was added to treated wells.

Cells were returned to the incubator for a further 2.5 h, then fixed and stained for total ubiquitin as described above.

## RNA-Seq

**RNA sample preparation.** HFFs in T25 flasks were serum starved for 24 h with DMEM containing 0.5% FBS. Treated flasks were simultaneously pre-stimulated for 24 h with 100 U/ml IFNγ (Bio-Techne, 285-IF-100). For infections, parasites were isolated by syringe lysis through a 30-gauge needle, passed through 5 μm filters, centrifuged and resuspended in 0.5% FBS DMEM. HFFs were infected in triplicate with 1 million parasites per flask, and 24 h post-infection, samples were collected by washing each flask ×1 in cold PBS, scraping in 2 ml PBS and transferring to an RNAse free tube. Samples were centrifuged at 2,000 rpm for 10 min, and then lysed in 600 μl RLT buffer. Lysates were homogenised using Qiashredders (Qiagen, 79656), then RNA was isolated using an RNeasy Mini Kit (Qiagen, 74104) according to manufacturer's instructions. mRNA libraries were prepared using the NEBNext Ultra II Directional PolyA mRNA kit (NEB, E7760L) with 100 ng of input, then sequenced on a NovaSeq (Illumina) using paired end 100 bp reads to a minimum depth of 30 million reads per sample.

**RNA-Seq data analysis.** RNA-seq data was quantified using STAR/RSEM [88] from within the nfcore/rnaseq [89] pipeline (version 3.4), against human and *Toxoplasma* transcriptomes (GRCh38, annotation release-95 from Ensembl and ToxoDB-59_TgondiiGT1 obtained from ToxoDB). Differential analysis was run across strain and IFNγ treatment groups using DESeq2 (1.36.0) [90], correcting for experimental batch effect in the model. Pairwise comparisons were run and an interactions analysis across the 2 experimental factors against uninfected samples and -IFNγ control group in each case. RSEM counts were imported using tximeta (1.14.1) to account for transcript length, and IHW (1.24.0) was used to control from multiple testing in the differential gene selection (<0.05 FDR). Shrunken log fold changes were calculated using type = "ashr".

RNASeq data have been deposited in the Gene Expression Omnibus (GEO) Database under accession number GSE230866.

## Supporting information

**S1 Fig. Verification of *Toxoplasma* knockout lines generated in this study. (A)** PCR verification of successful integration of repair cassettes at indicated gene loci. Primers used for verification are listed in S11 Data. **(B)** Strain genotype verification by restriction fragment length polymorphism (RFLP) of the SAG3 gene. **(C)** Quantification of plaque area after 10 days growth in HFFs. Results are shown as violin plots with median and quartiles from a minimum of 3 biological replicates. *p*-values were calculated by one-way analysis of variance (ANOVA) with Tukey's multiple comparison test. *, $p < 0.05$; ns, not significant. Source data for C can be found in S2 Data.
(TIF)

**S2 Fig. PCR verification of endogenously c-terminally tagged *Toxoplasma* lines generated in this study.** Primers used for verification are listed in S11 Data.
(TIF)

**S3 Fig. GRA57-HA partially co-localises with the PVM marker GRA3.** Scale bar represents 3 μm.
(TIF)

**S4 Fig. IFNγ stimulation of HFFs does not affect the localisation of GRA57-HA or GRA70-V5.** HFFs were pre-stimulated with 100 U/ml IFNγ for 24 h prior to infection,

fixation, and staining as in Fig 3F. Scale bar = 3 μm.
(TIF)

**S5 Fig. Survival of ΔGRA57 parasites is not rescued with exogenous supplementation of tryptophan.** HFF IFNγ restriction assays as in Fig 2, with the addition of 1 mM L-tryptophan to treated conditions simultaneous with IFNγ pre-stimulation. Untreated controls had 0.1 N NaOH added as a vehicle control. Host cells were infected in technical triplicate with the indicated parasite strains for 24 h at an MOI of 0.3, and then imaged live on a Cytation 5 plate reader. Total mCherry signal area per well was measured to determine parasite growth in IFNγ stimulated relative to unstimulated cells. Data displayed as mean survival + standard deviation from 2 biological replicates. Source data can be found in S3 Data.
(TIF)

**S6 Fig. GRA57 does not contribute to MYR-dependent translocation of host c-Myc. (A)** Representative IFA images. HFFs were infected for 24 h prior to fixation and staining for host c-Myc. **(B)** Quantification of nuclear translocation of host c-Myc. Median nuclear c-Myc signal in infected cells was measured in FIJI, with the median background c-Myc signal subtracted for each image. Median nuclear c-Myc fluorescence intensity for each strain was normalised to that of RHΔUPRT in each biological replicate. Data is shown as median with individual biological replicates overlayed. *p*-values were calculated by one-way analysis of variance (ANOVA) with Tukey's multiple comparison test. *, $p < 0.05$; **, $p < 0.01$; ns, not significant. Source data can be found in S6 Data.
(TIF)

**S7 Fig. GRA57 deletion does not affect formation of the IVN.** HFFs monolayers were infected with the indicated strains for 24 h prior to fixation and preparation for transmission electron microscopy (TEM).
(TIF)

**S8 Fig. Deletion of GRA57, GRA70, or GRA71 leads to reduced host ubiquitin recruitment in unstimulated cells. (A, B)** Recruitment of total host ubiquitin (FK2) to *Toxoplasma* vacuoles in **(A)** HFFs and **(B)** MEFs. Related to Fig 5A and 5B, data shows ubiquitin recruitment levels in unstimulated cells. Recruitment of ubiquitin was automatically counted using high-content imaging and analysis. *p*-values were calculated by paired two-sided *t* test. **(C)** Recruitment of total ubiquitin (FK2), K63-linked ubiquitin, linear ubiquitin (M1), and the E3 ligase RNF213 to *Toxoplasma* vacuoles in unstimulated HFFs. Related to Fig 5D, data shows specific ubiquitin linkage recruitment levels in unstimulated HFFs. Recruitment was automatically quantified for FK2, M1, and RNF213. K63 recruitment was manually scored, with minimum 100 vacuoles scored per condition. **(D)** Recruitment of K48-linked ubiquitin to *Toxoplasma* vacuoles in HFFs. Host cells were infected and stained for K48-linked ubiquitin as described in Fig 5, with automatic quantification of recruitment. *p*-values were calculated by paired two-sided *t* test. *, $p < 0.05$; **, $p < 0.01$; ***, $p < 0.001$; ****, $p < 0.0001$; ns, not significant. Source data can be found in S9 Data.
(TIF)

**S9 Fig. RHΔGRA57 vacuoles are targeted less frequently by host ubiquitin during the early stages of infection.** HFFs were infected as in Fig 5, and then fixed at the indicated time points post-infection. Data shown as mean of technical triplicate ± standard deviation. Source data can be found in S9 Data.
(TIF)

**S10 Fig. Inhibition of parasite egress in the first 3 h of infection does not restore ubiquitination or survival of ΔGRA57 vacuoles. (A)** Recruitment of total ubiquitin to *Toxoplasma* vacuoles at 3 h post-infection, with the addition of 5 μM Compound 2 at 30 min post-infection to inhibit parasite egress. HFFs were pre-stimulated with 100 U/ml IFNγ for 24 h, infected with indicated lines for 30 min prior to addition of Compound 2. HFFs were fixed at 3 h post-infection and stained for total ubiquitin. Recruitment was automatically counted using high-content imaging and analysis. **(B)** Parasite survival in IFNγ-stimulated HFFs at 3 h post-infection, with the addition of 5 μM Compound 2 at 30 min post-infection to inhibit parasite egress. Parasite numbers were quantified through automated high-content imaging, with survival calculated as the percentage of intracellular parasites in IFNγ-stimulated cells relative to the total in unstimulated cells. $p$-values were calculated by paired two-sided $t$ test. *, $p < 0.05$; **, $p < 0.01$; ***, $p < 0.001$; ****, $p < 0.0001$; ns, not significant. Source data can be found in S10 Data.
(TIF)

**S1 Raw Images. Raw images.**
(PDF)

**S1 Data. Data from CRISPR screens in RH and PRU parasites.** Raw read counts, normalised counts, guide L2FCs, gene L2FCs, $p$-values, and DISCO scores.
(XLSX)

**S2 Data. Raw data for plaque assays in HFFs.** Individual and median plaque area measurements.
(XLSX)

**S3 Data. Raw data for Cytation plate reader IFNγ survival assays. (A)** HFFs, **(B)** MEFs, and **(C)** HFFs +/− tryptophan supplementation. Total mCherry area +/− IFNγ per biological replicate.
(XLSX)

**S4 Data Co-immunoprecipitation and mass spectrometry results.**
(XLSX)

**S5 Data. Raw data for high-content imaging IFNγ survival assays in HFFs. (A)** *Toxoplasma* number, vacuole number, and mean vacuole area per well. **(B)** Survival percentages calculated from mean *Toxoplasma* number, vacuole number, or vacuole area +/− IFNγ per biological replicate.
(XLSX)

**S6 Data. Raw data for c-Myc nuclear translocation assays.** Nuclear c-Myc signal values and background signal values per biological replicate.
(XLSX)

**S7 Data. RNASeq of infected HFFs. Differential gene analysis and interaction analysis of human-aligned transcripts.** IFNγ treatment is denoted as yes or no. For the interaction analysis, sheet names represent the IFNγ effect for the first strain relative to the IFNγ effect for the second strain (i.e., WT|IFN_vs_dGRA57|IFN represents +IFNγ vs -IFNγ in the RHΔKu80 group relative to +IFNγ vs -IFNγ in the RHΔGRA57 group).
(XLSX)

**S8 Data. RNASeq of infected HFFs. Differential gene expression analysis of *Toxoplasma*-aligned transcripts.** IFNγ treatment is denoted as yes or no.
(XLSX)

**S9 Data. Raw data for vacuole recruitment assays. (A)** Percentage of vacuoles with ubiquitin recruitment per well in HFFs. **(B)** Mean percentage of vacuoles with ubiquitin recruitment in HFFs. **(C)** Percentage of vacuoles with ubiquitin recruitment per well in MEFs. **(D)** Mean percentage of vacuoles with ubiquitin recruitment in MEFs. **(E)** Percentage of vacuoles with total (FK2), K48-, K63-, or M1-linked ubiquitin or RNF213 recruitment in HFFs. **(F)** Percentage of vacuoles with ubiquitin recruitment at 1, 2, and 3 h post-infection per well in HFFs.
(XLSX)

**S10 Data. Raw data for Compound 2 egress inhibition assays. (A)** Percentage of vacuoles with ubiquitin recruitment +/− IFNγ and +/− Compound 2 per biological replicate. **(B)** Survival percentages at 3 h post-infection calculated from number of intracellular *Toxoplasma* per biological replicate.
(XLSX)

**S11 Data. Primer sequences used in this work.**
(XLSX)

**S12 Data. Opera Phenix image acquisition parameters and Harmony analysis sequence used for automated ubiquitin recruitment analysis.**
(DOCX)

**S13 Data. Antibodies used for immunofluoresence assays.**
(DOCX)

## Acknowledgments

We thank all members of the Treeck laboratory as well as Barbara Clough and Eva Frickel for critical discussions and Stephanie Nofal for critically reading the manuscript. We thank Michael Howell, Rachael Instrell, and Becky Saunders (High-Throughput Screening Science Technology Platform, The Francis Crick Institute, London, United Kingdom) for assistance with sgRNA library preparation. We thank members of the Advanced Sequencing, High Throughput Screening, Electron Microscopy, Proteomics and Cell Services Science Technology Platforms at the Francis Crick Institute for support. We thank Bishara Marzook for assistance with the Cytation 5 plate imager, Jean Francois Dubremetz for providing the GRA3 antibody, Felix Randow and Ana Crespillo-Casado for providing the MEFs, and Matthew Cottee for assistance with protein structural analysis. We acknowledge ToxoDB (http://toxodb.org/) for providing an invaluable resource that made this work possible.

## Author Contributions

**Conceptualization:** Eloise J. Lockyer, Francesca Torelli, Moritz Treeck.

**Formal analysis:** Eloise J. Lockyer, Ok-Ryul Song, Steven Howell, Anne Weston, Philip East.

**Funding acquisition:** Moritz Treeck.

**Investigation:** Eloise J. Lockyer.

**Methodology:** Eloise J. Lockyer, Francesca Torelli, Simon Butterworth, Ok-Ryul Song, Steven Howell, Anne Weston.

**Project administration:** Moritz Treeck.

**Software:** Simon Butterworth.

**Supervision:** Moritz Treeck.

**Visualization:** Eloise J. Lockyer, Francesca Torelli.

**Writing – original draft:** Eloise J. Lockyer, Moritz Treeck.

**Writing – review & editing:** Eloise J. Lockyer, Francesca Torelli, Simon Butterworth, Moritz Treeck.

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
