## [Editor Report · Decision Letter 0]

27 Apr 2023

Dear Dr. Treeck, 

Thank you for submitting your manuscript entitled "A heterotrimeric complex of Toxoplasma proteins promotes parasite survival in interferon gamma stimulated human cells" for consideration as a Research Article by PLOS Biology.

Your manuscript has now been evaluated by the PLOS Biology editorial staff, as well as by an academic editor with relevant expertise, and I am writing to let you know that we would like to send your submission back for re-review to the same reviewers from Review Commons.

Once your full submission is complete, your paper will undergo a series of checks in preparation for peer review. After your manuscript has passed the checks it will be sent out for review. To provide the metadata for your submission, please Login to Editorial Manager (https://www.editorialmanager.com/pbiology) within two working days, i.e. by Apr 29 2023 11:59PM.

Kind regards,

Paula

---

Senior Editor

PLOS Biology

---

## [Decision Letter · Decision Letter 1]

2 Jun 2023

Dear Dr. Treeck,

Thank you for your patience while we considered your revised manuscript "A heterotrimeric complex of Toxoplasma proteins promotes parasite survival in interferon gamma stimulated human cells" for publication as a Research Article at PLOS Biology. This revised version of your manuscript has been evaluated by the PLOS Biology editors, the Academic Editor and the original reviewers.

Based on the reviews, we are likely to accept this manuscript for publication, provided you satisfactorily address the remaining points raised by the reviewers. In particular, to address the concern from reviewer #1 regarding the limited mechanism, we consider that you could add this limitation in the discussion, explaining what you would need to do to find the mechanistic insights and that this would require too much work for this manuscript.

Please also make sure to address the following data and other policy-related requests.

1. DATA POLICY:

A) Supplementary files (e.g., excel). Please ensure that all data files are uploaded as 'Supporting Information' and are invariably referred to (in the manuscript, figure legends, and the Description field when uploading your files) using the following format verbatim: S1 Data, S2 Data, etc. Multiple panels of a single or even several figures can be included as multiple sheets in one excel file that is saved using exactly the following convention: S1_Data.xlsx (using an underscore).

B) Deposition in a publicly available repository. Please also provide the accession code or a reviewer link so that we may view your data before publication.

Regardless of the method selected, please ensure that you provide the individual numerical values that underlie the summary data displayed in the following figure panels as they are essential for readers to assess your analysis and to reproduce it: Figures 1BCD, 2CD, 3AB, 4ABC, 5BCD, and Supplementary Figures S1C, S5, S6B, S8ABCD, S9, S10AB.

**Please also ensure that figure legends in your manuscript include information on where the underlying data can be found, and ensure your supplemental data file/s has a legend.**

We require the original, uncropped and minimally adjusted images supporting all blot and gel results reported in an article's figures or Supporting Information files. We will require these files before a manuscript can be accepted so please prepare and upload them now. We require this for Figures 2B, 3D, S1AB, S2.

Please carefully read our guidelines for how to prepare and upload this data: https://journals.plos.org/plosbiology/s/figures#loc-blot-and-gel-reporting-requirements

We expect to receive your revised manuscript within two weeks.

*Published Peer Review History*

*Press*

Sincerely,

Paula

---

Senior Editor,

pjaureguionieva@plos.org,

PLOS Biology

Reviewer remarks:

Reviewer #1: The authors have addressed most of the comments raised by the reviewers and have incorporated revisions to their manuscript. They have added new data, specifically regarding the ubiquitin phenotype, to strengthen their conclusions. Additionally, they have included explanations and new discussion points in the text. In doing so, the authors have likely enhanced the quality and validity of their manuscript, which also nicely confirms the data published by Krishnamurthy et al., mBio 2023. However, although this article addresses an important aspect of human cell autonomous immunity against Toxoplasma, the exact host defense mechanisms involved remain to be identified, as well as a description of the mechanism by which these effectors oppose IFNgamma-induced cell defenses, which considerably limits its impact in the fields of parasitology and immunology.

Reviewer #2: the revised MS addresses my main critiques adequately. This study is of substantial interest to the parasitology field

Reviewer #3: In my opinion, the authors have addressed the issues by all reviewers adequately and have performed necessary experimental work in addition to improving the manuscript or providing a rebuttal. Looking back at my comment regarding figure 4A, I myself fail to understand what I meant. I am sorry about this; I guess it was something about a completely different figure. I understand that some of the suggested experiments are too much work for this paper. Some of the issues have already been indirectly addressed in the paper by Krishnamurthy et al. 2023. I am looking forward to seeing more work from this group.

---

## [Editor Report · Decision Letter 2]

16 Jun 2023

Dear Dr Treeck,

Thank you for the submission of your revised Research Article "A heterotrimeric complex of Toxoplasma proteins promotes parasite survival in interferon gamma stimulated human cells" for publication in PLOS Biology. On behalf of my colleagues and the Academic Editor, Kami Kim, I am pleased to say that we can in principle accept your manuscript for publication, provided you address any remaining formatting and reporting issues. These will be detailed in an email you should receive within 2-3 business days from our colleagues in the journal operations team; no action is required from you until then. Please note that we will not be able to formally accept your manuscript and schedule it for publication until you have completed any requested changes.

PRESS

Sincerely, 

Paula

---

Senior Editor

PLOS Biology
